# Oncogenic BRAF induces whole-genome doubling through suppression of cytokinesis

Revati Darp[1,2], Marc A. Vittoria[3], Neil J. Ganem [3] & Craig J. Ceol [1,2✉]

Melanomas and other solid tumors commonly have increased ploidy, with near-tetraploid karyotypes being most frequently observed. Such karyotypes have been shown to arise through whole-genome doubling events that occur during early stages of tumor progression. The generation of tetraploid cells via whole-genome doubling is proposed to allow nascent tumor cells the ability to sample various pro-tumorigenic genomic configurations while avoiding the negative consequences that chromosomal gains or losses have in diploid cells. Whereas a high prevalence of whole-genome doubling events has been established, the means by which whole-genome doubling arises is unclear. Here, we find that BRAF[V600E], the most common mutation in melanomas, can induce whole-genome doubling via cytokinesis failure in vitro and in a zebrafish melanoma model. Mechanistically, BRAF[V600E] causes decreased activation and localization of RhoA, a critical cytokinesis regulator. BRAF[V600E] activity during G1/S phases of the cell cycle is required to suppress cytokinesis. During G1/S, BRAF[V600E] activity causes inappropriate centriole amplification, which is linked in part to inhibition of RhoA and suppression of cytokinesis. Together these data suggest that common abnormalities of melanomas linked to tumorigenesis – amplified centrosomes and whole-genome doubling events – can be induced by oncogenic BRAF and other mutations that increase RAS/MAPK pathway activity.

[1] University of Massachusetts Chan Medical School, Program in Molecular Medicine, Worcester, MA, USA. [2] University of Massachusetts Chan Medical School, Department of Molecular, Cellular and Cancer Biology, Worcester, MA, USA. [3] Departments of Pharmacology and Experimental Therapeutics and Medicine, Division of Hematology and Oncology, Boston University School of Medicine, Boston, MA, USA. ✉email: Craig.Ceol@umassmed.edu

ncreased ploidy is a common feature of solid tumors. The most frequently observed increased karyotypes approach tetraploidy, which led to the hypothesis that such 'near-tetraploid' tumors had undergone a whole-genome doubling (WGD) event during tumor progression and subsequently experienced a small net loss of chromosomes[1,2]. Recent bioinformatic analyses support this hypothesis, showing that WGD events are prevalent in a diverse set of solid tumors, and nearly 37% of all solid tumors measured, including 40% of melanomas, experienced at least one WGD event in their progression[3,4]. Based on these analyses, WGD frequently occurs early in tumor formation, and the presence of tetraploid cells in some pre-cancerous lesions, such as Barrett's esophagus and lesions of the cervix and kidney, suggests that WGD may even precede frank tumor formation in some tissues[5–8] Tetraploidy was also observed in hyperplastic lesions of the pancreas[9], in localized prostate cancer[10–12] and some colon adenomas[13,14], and for certain malignancies, such as oral tumors[15], tetraploidy is a strong predictor of malignant transformation. Additionally, in established cancers from many tissue types WGD is a predictor of poor clinical outcome[16].

Tetraploidy has been experimentally linked to tumorigenesis. Viral-induced cell fusion has been shown to enhance the transformation and tumor-forming capabilities of different cell types[17–20]. Additionally, in mouse mammary epithelial cells that were made tetraploid through treatment with the actin filament poison dihydrocytochalasin B, tetraploid cells were able to form tumors in mice whereas their isogenic diploid counterparts were not[21]. In support of a role for WGD in tumorigenesis, deep sequencing of tumor samples has shown WGD to be an early event in non-small cell lung cancer, medulloblastoma and other tumor types[1,2,22,23].

There are different and mutually inclusive ways in which tetraploidy could contribute to tumorigenesis. First, tetraploidy can enable cells to become tolerant to the negative consequences of chromosome gains, losses, gene deletions, and inactivating mutations[24–30]. Hence, tetraploidy is likely to allow tumor cells to withstand a higher incidence of mutations, thereby increasing the probability of adaptive changes. Second, tetraploid cells have an increased rate of chromosome missegregation[31–33], thus increasing the possibility that a developing tumorigenic clone will accumulate and tolerate the mutations needed for its progression to a malignant state[34]. Thirdly, proliferating tetraploid cells are genetically unstable and can facilitate tumor progression by giving rise to aneuploidy, a known hallmark of cancer[35].

Melanomas are a tumor type in which WGD is prevalent[4]. Although molecular genetic analyses have provided great insights into the genes that are involved in melanoma, very little is known about the process by which melanocytes with these lesions become tumorigenic, and whether any mutations underlie WGD in tumors is unclear. We examined melanocytes in zebrafish strains that are predisposed to melanoma and discovered an abundance of binucleate, tetraploid melanocytes. Tetraploidy was caused by expression of $BRAF^{V600E}$, which increases RAS/MAPK-pathway activity and is commonly found in human melanomas. Using an in vitro model combined with live imaging, flow cytometry and immunofluorescence approaches, we found that $BRAF^{V600E}$ generated tetraploid cells via cytokinesis failure and reduced activity of the small GTPase RhoA, which is critical for cytokinesis[36]. We also show that $BRAF^{V600E}$ activity causes inappropriate centrosomal amplification, which is linked in part to the inhibition of RhoA and suppression of cytokinesis. Additionally, we show that zebrafish melanomas have a tetraploid karyotype and tumor-initiating cells in the zebrafish are tetraploid. These data collectively suggest that $BRAF^{V600E}$-induced WGD occurs and has a role in tumor formation.

## Results

**$BRAF^{V600E}$ causes melanocytes in zebrafish to be tetraploid and binucleate.** In this and previous studies, we used a zebrafish model of melanoma which combined melanocyte-lineage expression of human $BRAF^{V600E}$ with an inactivating mutation in the endogenous zebrafish p53 gene[37–39]. Animals of this genotype, $Tg(mitfa:BRAF^{V600E})$; $p53(lf)$, develop melanomas that have histopathological and molecular features similar to those of human melanomas. To determine if tumors arising in this model exhibited ploidies consistent with having undergone a WGD event, we harvested tumors from $Tg(mitfa:BRAF^{V600E})$; $p53(lf)$ animals and quantified DNA content. The ploidy of these zebrafish melanomas was predominantly 4N and higher (Fig. 1A), indicating that WGD is likely a feature of this model. The analyzed tumors displayed a small fraction of 2N cells, which we speculate were admixed stromal cells.

To investigate when the WGD event could occur, we began by examining melanocytes from $Tg(mitfa:BRAF^{V600E})$; $p53(lf)$ animals. We reasoned that closer analyses of epidermal melanocytes in the $Tg(mitfa:BRAF^{V600E})$ strains might reveal the basis of the observed tetraploidy and provide insight into early cellular events that occur in melanoma tumorigenesis. To this end we developed assays to quantify and determine cell biological characteristics of these melanocytes. To quantify dorsal epidermal melanocytes, we treated fish with epinephrine then plucked and fixed scales to which these melanocytes are attached. As the number of melanocytes per scale depends on the size of the scale, we obtained a normalized melanocyte density measurement. Melanomas arise from dorsal regions of these zebrafish, and we found that the scale-associated epidermal melanocytes in these dorsal regions were larger in size and fewer in number than those of wild-type zebrafish (Fig. 1B, C, Supplementary Fig. 1A). This was due to $BRAF^{V600E}$ expression, as $Tg(mitfa:BRAF^{V600E})$ melanocytes were also larger and fewer in number, whereas $p53(lf)$ melanocytes were similar to those of wild-type zebrafish. Previously, injection of $BRAF^{V600E}$ had been shown to cause nevus-like proliferations of melanocytes in zebrafish[38], and we also showed nevus-like melanocyte proliferations can arise in the $Tg(mitfa:BRAF^{V600E})$; $p53(lf)$ strain[37]. However, our current characterization of melanocytes in strains stably expressing the $Tg(mitfa:BRAF^{V600E})$ transgene indicates that, aside from a few melanocytes that clonally proliferate, $BRAF^{V600E}$ expression primarily results in a reduced number and increased size of melanocytes. Cell size increases can be caused by increased ploidy, which could be reflected in a larger nuclear size[40]. To determine if $BRAF^{V600E}$ expression caused nuclear enlargement, we stained for the melanocyte nuclear protein Mitfa. Most nuclei in large $Tg(mitfa:BRAF^{V600E})$ melanocytes were similar in size to those of wild-type melanocytes; however, melanocytes in $Tg(mitfa:BRAF^{V600E})$ contained two nuclei (Fig. 1D, E). Melanocytes from $Tg(mitfa:BRAF^{V600E})$; $p53(lf)$ animals were similarly binucleate, whereas $p53(lf)$ melanocytes had one nucleus (Fig. 1E, Supplementary Fig. 1B), indicating that the binuclearity is associated with $BRAF^{V600E}$ expression. Whereas binucleate dorsal epidermal melanocytes are rare in wild-type animals, binucleate stripe-associated dermal melanocytes are more commonly observed[41,42], suggesting that mechanisms that coordinate nuclear and cellular divisions may be particularly prone to regulation in zebrafish melanocytes.

To determine if the binuclearity we observed was uniquely associated with $BRAF^{V600E}$ or was caused by Ras/BRAF pathway overactivity in general, we stained scale-associated melanocytes expressing a common oncogenic variant of NRAS, mutations in which are present in about 28% of human melanomas[43]. Melanocytes from animals expressing an $NRAS^{Q61L}$ oncogene that is commonly found in human melanomas were also binucleate (Fig. 1E, Supplementary Fig. 1C)[44], and binucleation has also been

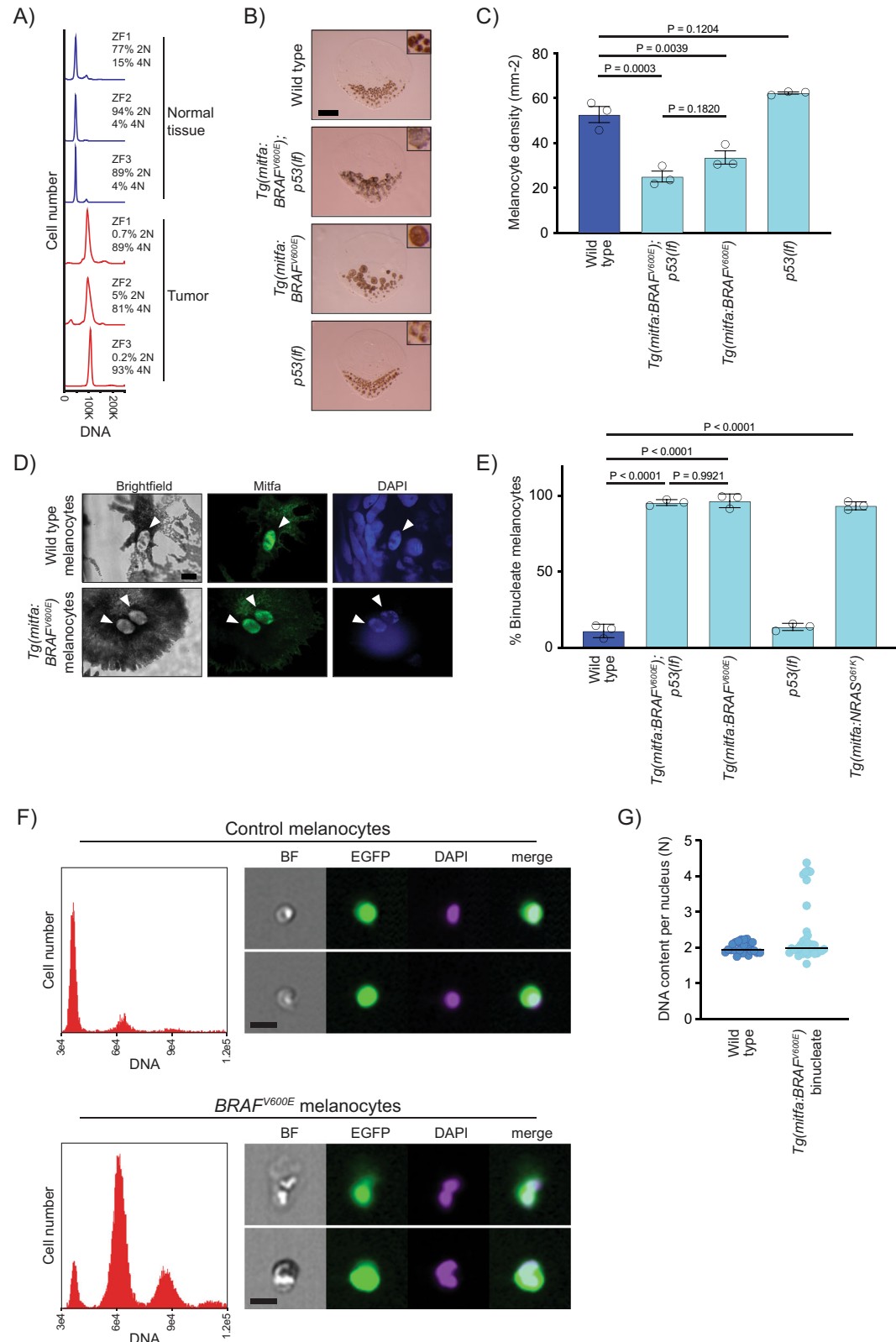

observed in zebrafish melanocytes that express an oncogenic variant of *HRAS*[45]. Together these data indicate that binucleate cells arise from overactivation of RAS/MAPK signaling.

To examine ploidy of *Tg(mitfa:BRAF^{V600E})* melanocytes, flow cytometry and DNA densitometry were performed. Zebrafish melanocytes retain melanin pigment, so an *mitfa:EGFP* transgene and *albino* mutation were introduced so that melanocytes could be

reliably identified by GFP-positivity and characterized without melanin spectral interference. Flow cytometry showed that *Tg(mitfa:BRAF^{V600E})* melanocytes were predominantly tetraploid, with small fractions of diploid and octoploid cells observed (Fig. 1F). Flow cytometry also confirmed the binucleate nature of these cells (Supplementary Fig. 1D). DNA densitometry of fixed samples found that most nuclei in binucleate *Tg(mitfa:BRAF^{V600E})*

**Fig. 1 $BRAF^{V600E}$ causes melanocytes in zebrafish to be tetraploid. A** DNA content of normal and tumor tissue from $Tg(mitfa:EGFP)$; $Tg(mitfa:BRAF^{V600E})$; $p53(lf)$; $alb(lf)$ zebrafish. ZF1, ZF2, and ZF3 are three different animals. **B** Scales from wild-type, $Tg(mitfa:BRAF^{V600E})$;$p53(lf)$, $Tg(mitfa:BRAF^{V600E})$ and $p53(lf)$ strains. Melanin pigment is dispersed throughout the cytoplasm of zebrafish melanocytes, revealing markedly different cell sizes. Scale bar = 250 μm, insets are at same scale as one another. **C** Quantification of melanocyte densities per scale of wild-type, $Tg(mitfa:BRAF^{V600E})$;$p53(lf)$, $Tg(mitfa:BRAF^{V600E})$ and $p53(lf)$ strains. $N = 5$ for three animals of each genotype; the mean density of each animal is plotted. One-way ANOVA with Tukey's multiple comparisons test. Error bars represent mean ± SEM. **D** Images from brightfield (left), anti-Mitfa (middle) and DAPI (right) staining of a single wild-type (top) or $Tg(mitfa:BRAF^{V600E})$ (bottom) epidermal melanocyte. Only the melanocyte nuclei stain positively for Mitfa. White arrowheads indicate nuclei within a single melanocyte. Scale bar = 5 μm. Representative cells quantified in **E** are shown. **E** Mean percent binucleate cells as determined by anti-Mitfa staining of pigmented melanocytes. $N = 3$ experiments examining in total wild type = 636, $Tg(mitfa:BRAF^{V600E})$;$p53(lf)$ = 458, $Tg(mitfa:BRAF^{V600E})$ = 454, $p53(lf)$ = 538, and $Tg(mitfa:NRAS^{Q61K})$ = 122 melanocytes. One-way ANOVA with Tukey's multiple comparisons test. Error bars represent mean ± SEM. **F** Flow cytometry and DNA content analysis of control $Tg(mitfa:EGFP)$; $alb(lf)$ and $Tg(mitfa:EGFP)$; $Tg(mitfa:BRAF^{V600E})$; $alb(lf)$ melanocytes with brightfield, EGFP and DAPI images of single melanocytes. **G** DNA content analysis of wild-type and $Tg(mitfa:BRAF^{V600E})$ melanocyte nuclei by confocal densitometry. $N = 25$ nuclei for wild type and $N = 35$ for $Tg(mitfa:BRAF^{V600E})$. Bars represent median.

melanocytes had a 2N DNA content (Fig. 1G). Therefore, expression of $BRAF^{V600E}$ causes melanocytes in zebrafish to become tetraploid as a result of having two nuclei, each with a 2N DNA content.

**$BRAF^{V600E}$ binucleate, tetraploid cells arise via failure of cytokinesis**. To determine how $BRAF^{V600E}$ generates binucleate, tetraploid cells and to recapitulate the phenotype we observed in our zebrafish model, we developed an in vitro system suitable for mechanistic analyses. This system is based on a single-cell clone we created in which $BRAF^{V600E}$ was inducibly expressed in RPE-1 FUCCI cells at a level similar to that of endogenous BRAF (Fig. 2A, B). RPE-1 cells were chosen for this analysis because they are non-transformed, hTERT-immortalized, and have a stable, diploid karyotype. We combined DNA content analysis with the fluorescent ubiquitin-based cell cycle indicator (FUCCI) reporter system to enable abnormal tetraploid cells in the G1 phase of the cell cycle (Cdt-mCherry-positive) to be distinguished from normal tetraploid cells in the G2 or M phases of the cell cycle (Geminin-GFP-positive)[46,47]. In these cells, synchronization was performed by serum starvation, then after serum addition and progression through mitosis $BRAF^{V600E}$ was induced using doxycycline, and cells were synchronized again with a thymidine block. Following release from thymidine synchronization and progression through mitosis, G1 tetraploids were measured as Cdt-mCherry-positive cells with a 4N DNA content (Fig. 2C, Supplementary Fig. 2A). Expression of $BRAF^{V600E}$ caused a nearly three-fold increase in the percentage of G1 tetraploid cells (Fig. 2D). These cells were CyclinD1-positive, confirming that they were in G1 and not G2 cells that had dysregulated Cdt-mCherry expression (Supplementary Fig. 2B). Expression of wild-type or kinase-dead $BRAF$ showed no similar increase in G1 tetraploids (Fig. 2D). Using a single-cell clone in which $BRAF^{V600E}$ was inducibly expressed in RPE-1 H2B-GFP cells, live-cell imaging was used to investigate whether the G1 tetraploids were binucleate and, if so, how they arose. Indeed, after $BRAF^{V600E}$ induction and release from synchronization, binucleate cells were observed in the following G1 phase of the cell cycle (Fig. 2E, F). These cells arose through failure of cytokinesis characterized by the formation then regression of the cytokinetic cleavage furrow. Modest increases in other mitotic defects, including lagging chromosomes, chromosome bridges and micronuclei, were observed although none of these increases was statistically significant (Supplementary Fig. 2C–E). Mitotic duration was not affected, even in cells that had undergone cytokinesis failure (Supplementary Fig. 2F). Together these data indicate that oncogenic $BRAF^{V600E}$ causes WGD and the formation of binucleate, tetraploid cells by impairment of cytokinesis rather than cell fusion or other means.

**$BRAF^{V600E}$ causes cytokinesis failure by reducing the localization and function of RhoA**. To understand how $BRAF^{V600E}$ inhibits cytokinesis, we investigated the localization and function of proteins that are involved in mitotic exit and the cytokinetic process. Polo-like kinase I (PLK1) helps to initiate cytokinesis[48], and it has been shown to interact with CRAF in G2/M[49]. We tested whether cytokinesis failure was dependent on PLK1 and found that an inhibitor of PLK1 did not affect BRAFV600E tetraploid formation (Supplementary Fig. 3A). MPS1/TTK1, a kinase which is activated by BRAFV600E and promotes activation of the mitotic spindle checkpoint[50], was not activated in response to BRAFV600E expression in RPE-1 cells and thus not likely involved in BRAFV600E-driven cytokinetic failure (Supplementary Fig. 3B). Other proteins that have been extensively characterized as being critical during cytokinesis, mainly in contractile ring and cleavage furrow formation, are RhoA, a member of the RhoGTPase family and its scaffold protein Anillin[36,51]. Following anaphase Anillin is required to maintain the assembly of cytokinetic furrow components at the equatorial cell cortex. In cells expressing $BRAF^{V600E}$, Anillin staining was greatly reduced (Fig. 3A, B). Anillin localization is regulated by RhoA[52], which activates and coordinates several downstream events in the cytokinetic process. RhoA is spatiotemporally activated and accumulates at the equatorial cell cortex in anaphase and during cytokinesis. This accumulation is both necessary and sufficient for cytokinesis to proceed[53]. Similar to Anillin, RhoA localization to the cell equator was reduced in $BRAF^{V600E}$-expressing cells (Fig. 3C, D). RhoA reduction in $BRAF^{V600E}$-expressing cells was dependent on increased MAPK signaling because treatment with the MEK inhibitor trametinib or ERK inhibitor SCH772984 restored RhoA localization (Fig. 3C, D). Since RhoA localization is promoted by its activation at the equatorial cell cortex[54], we quantified the levels of active, GTP-bound RhoA in $BRAF^{V600E}$-expressing cells. Levels of GTP-bound RhoA were reduced as a consequence of $BRAF^{V600E}$ expression (Fig. 3E, F). $BRAF^{V600E}$-dependent reduction in GTP-bound RhoA has also been observed in immortalized melanocyte Mel-ST cells, indicating relevance to melanocyte biology[55]. If $BRAF^{V600E}$ acts to reduce function of RhoA, then an increase in RhoA function would be predicted to suppress the effect of $BRAF^{V600E}$ on the formation of binucleate, tetraploid cells. This occurred, as treatment of $BRAF^{V600E}$-expressing cells with RhoA activators LPA or S1P reduced the formation of tetraploid cells (Fig. 3G). Furthermore, expression of the $RHOA^{Q61L}$ activated variant suppressed $BRAF^{V600E}$-induced tetraploidy (Fig. 3H). These data indicate that $BRAF^{V600E}$ reduces the activity of RhoA and its downstream effector Anillin, which underlies the failure of cytokinesis and the formation of binucleate, tetraploid cells.

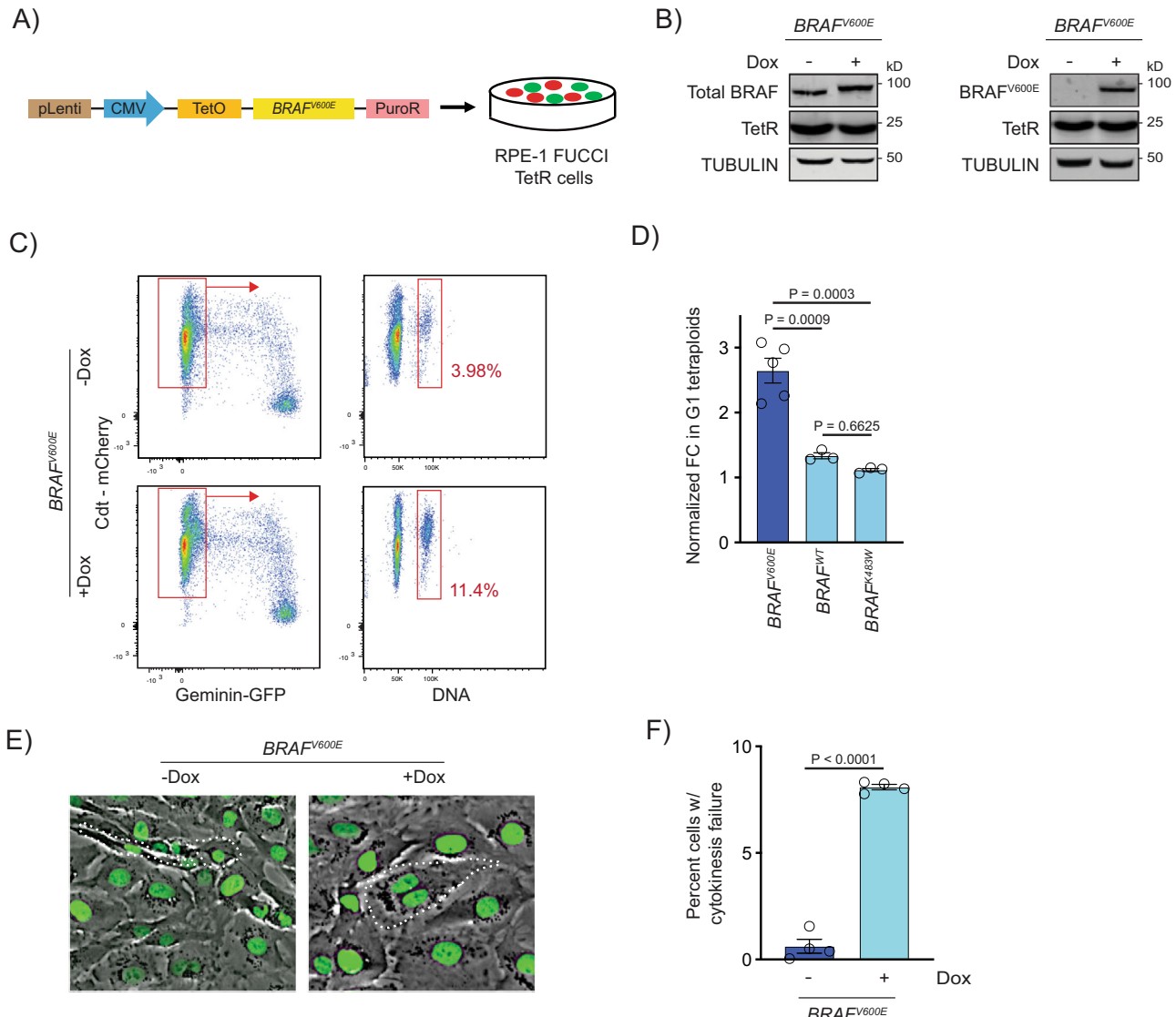

**Fig. 2 BRAF^{V600E}-induced binucleate, tetraploid cells arise via cytokinesis failure. A** Generation of *BRAF^{V600E}*-expressing RPE-1 FUCCI cell lines with a lentiviral-based doxycycline-inducible vector. **B** Western blot showing inducible expression of BRAF^{V600E} (+Dox) in RPE-1 FUCCI cells using a BRAF and BRAF^{V600E}-specific antibody. Expression of the Tet repressor protein is shown. TUBULIN was used as the loading control. A representative of three independent biological replicates is shown. **C** Flow cytometry plots of control (−Dox) and *BRAF^{V600E}*-expressing (+Dox) RPE-1 FUCCI cells. Tetraploid cells accumulating in G1 were quantified based on Cdt1-mCherry positivity and Hoechst incorporation. Percentages of G1 tetraploid cells in control and *BRAF^{V600E}*-expressing cultures are indicated. **D** Fold change (FC) in G1 tetraploids normalized to the control (−Dox) are shown for *BRAF^{V600E}*, *BRAF^{WT}* and *BRAF^{K483W}* (kinase-dead) -expressing RPE-1 FUCCI cell lines. $N = 5$ independent experiments for *BRAF^{V600E}*, $N = 3$ for *BRAF^{WT}* and $N = 3$ for *BRAF^{K483W}*. One-way ANOVA with Tukey's multiple comparisons test. Error bars represent mean ± SEM. **E** Merged phase contrast and GFP photomicrographs of H2B-GFP expressing control (−Dox) and *BRAF^{V600E}*-expressing (+Dox) RPE-1 cells that have recently undergone mitosis. White dotted lines indicate 2 cells with 1 nucleus each that have separated following a successful cytokinesis (−Dox) and 1 cell with 2 nuclei that has failed cytokinesis (+Dox). **F** Percentages of cells exhibiting cytokinesis failure in H2B-GFP RPE-1 control (−Dox) and *BRAF^{V600E}*-expressing (+Dox) cells. $N = 4$ independent experiments examining -Dox = 934 and +Dox = 568 cells. Unpaired Student's *t* test. Error bars represent mean ± SEM.

**BRAF^{V600E} and MAPK pathway activity is required during late G1 and early S phases for generating tetraploids.** BRAF^{V600E} acts in G1/S to promote cell cycle progression, yet some reports have suggested that MAPK activity is important during mitosis[56–58]. To determine when BRAF^{V600E} and MAPK signaling is required to generate tetraploids, we treated *BRAF^{V600E}*-expressing cells with the BRAF^{V600E} inhibitor vemurafenib, MEK inhibitor trametinib and ERK inhibitor SCH772984 at various points in the cell cycle and measured whether the inhibitors suppressed tetraploid formation (Fig. 4A). The ability of inhibitors to reduce downstream MAPK activity was confirmed by western blot of phosphorylated ERK (Supplementary Fig. 4A, B).

Additionally, we also confirmed that treatment with inhibitors at the concentrations used did not cause cell cycle arrest and did not substantially impact growth kinetics, cell cycle progression or cell viability (Supplementary Fig. 4C–E). Treatment with inhibitors throughout the cell cycle suppressed formation of *BRAF^{V600E}*-induced tetraploids (Fig. 4A). By contrast, treatment during the S/G2/M phases had little effect on tetraploid formation. Treatment during G1 and specifically during late G1 and early S suppressed formation of tetraploids. This suppression occurred with each of the three inhibitors tested as well as the 'paradox-breaking' BRAF^{V600E} inhibitors PLX7904 and PLX8394 (Supplementary Fig. 4F), which inhibit BRAF^{V600E} while not simultaneously

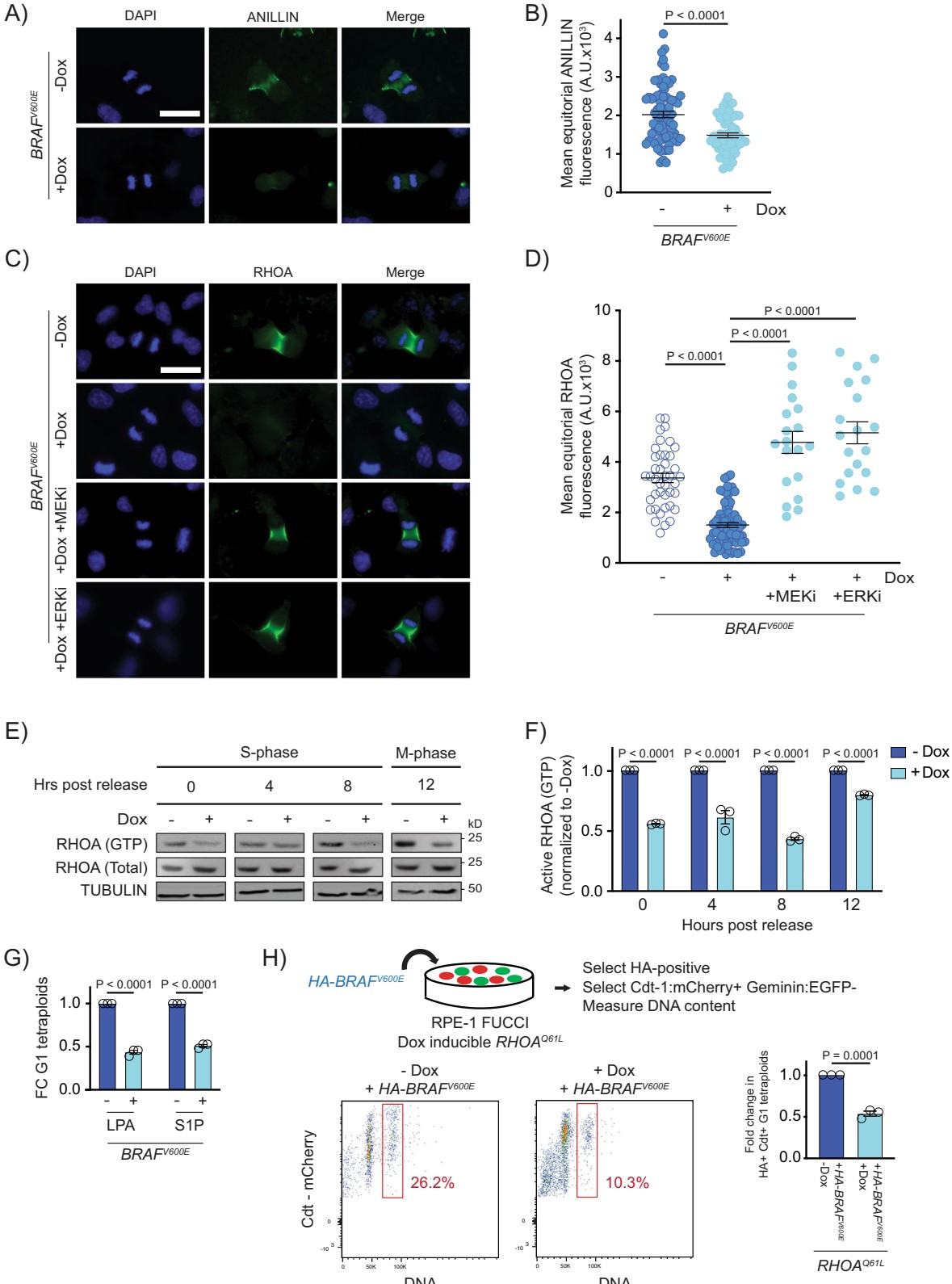

activating downstream MAPK activity like vemurafenib can[59–61]. Thus, BRAF$^{V600E}$, and MAPK signaling in general, act during late G1 and early S phases to ultimately cause inhibition of RhoA and failure of cytokinesis.

**Rac1 is activated by BRAF$^{V600E}$ and contributes to RhoA downregulation.** To understand how BRAF$^{V600E}$ and MAPK

signaling activity during G1/S could lead to downregulation of RhoA and failed cytokinesis, we considered regulators of cytokinesis that are active during G1 or S phases and whose dysregulation could impair cytokinesis. The small GTPase Rac1 is a negative regulator of cytokinesis that inhibits function of the contractile ring[53]. Rac1 is active throughout the cell cycle except during a small window in mitosis, with its nadir of activity during

**Fig. 3 BRAF$^{V600E}$ causes cytokinesis failure by reducing the localization and function of RhoA. A** DAPI and anti-ANILLIN staining in control (−Dox) and *BRAF$^{V600E}$*-expressing (+Dox) anaphase RPE-1 cells. Images are maximum intensity projections of z-stacks. Scale bar = 7.5 μM. **B** Mean ANILLIN fluorescence intensity at the equator of control and *BRAF$^{V600E}$*-expressing anaphase RPE-1 cells. N = 79 cells for −Dox and N = 53 for +Dox. Unpaired Student's *t* test. Error bars represent mean ± SEM. **C** DAPI and anti-RHOA staining in -*BRAF$^{V600E}$* (−Dox) cells, *BRAF$^{V600E}$*-expressing (+Dox) cells, and *BRAF$^{V600E}$*-expressing (+Dox) RPE-1 cells treated with MEKi or ERKi. Drugs were added coincident with Dox administration. Images are maximum intensity projections of z-stacks (0.20 μM). Scale bar = 7.5 μM. **D** Mean RHOA fluorescence intensity at the equator of control RPE-1 cells, *BRAF$^{V600E}$*-expressing RPE-1 cells, and *BRAF$^{V600E}$*-expressing RPE-1 cells treated with MEKi or ERKi. N = 41 cells for −Dox, N = 68 for +Dox, N = 19 for +Dox +MEKi, and N = 19 for +Dox +ERKi. Brown-Forsythe and Welch one-way ANOVA with Dunnett's multiple comparisons test. Error bars represent mean ± SEM. **E** Western blot analysis of immunoprecipitated RHOA-GTP from control (−Dox) and *BRAF$^{V600E}$*-expressing (+Dox) RPE-1 cell lysates at different time points post thymidine release. Total RHOA protein and alpha tubulin were used as a controls. A representative of three independent biological replicates is shown. **F** Western blot quantification of immunoprecipitated RHOA-GTP levels from control (−Dox) and *BRAF$^{V600E}$*-expressing (+Dox) RPE-1 cell lysates at different time points post thymidine release. Samples were normalized to the −Dox condition. N = 3 independent experiments. Unpaired Student's *t* test. Error bars represent mean ± SEM. **G** Fold change (FC) in G1 tetraploid RPE-1 FUCCI cells following addition of RHOA activators. LPA (1 μM) and S1P (1 μM) were added coincident with Dox administration. Fold changes in G1 tetraploids relative to the control (+Dox no drug) are shown. N = 3 independent experiments. Unpaired Student's *t* test. Error bars represent mean ± SEM. **H** G1 tetraploid generation following expression of HA-tagged-BRAF$^{V600E}$ in *RHOA$^{Q61L}$*-inducible cells. Experimental design (top): RPE-1 FUCCI cells with Dox inducible *RHOA$^{Q61L}$* were transiently transfected with an HA-tagged-BRAF$^{V600E}$-expressing construct and selected accordingly. G1 tetraploids were quantified (bottom left and middle) by gating HA-positive, Cdt-1:mCherry-positive cells with increased Hoechst incorporation. Fold change in G1 tetraploids (bottom right), normalized to control (−Dox) cells. N = 3 independent experiments. Unpaired Student's *t* test. Error bars represent mean ± SEM.

anaphase and early telophase[62]. Loss of Rac1 activity promotes cytokinesis[63], and overactivation of Rac1 impairs RhoA localization to the equatorial cell cortex[64] and causes cytokinesis failure leading to formation of binucleate cells[65]. To investigate whether BRAF$^{V600E}$ affects Rac1 activity, we performed ELISA assays to detect active, GTP-bound Rac1 at various points following BRAF$^{V600E}$ expression and release from synchronization. As compared to control cells, GTP-bound Rac1 levels in *BRAF$^{V600E}$*-expressing cells were higher upon release from synchronization and thereafter (Fig. 4B). To determine if higher Rac1 activity contributes to the *BRAF$^{V600E}$*-induced formation of binucleate cells, we treated cells with the Rac1 inhibitors NSC2366 and EHT1864 and measured G1 tetraploid formation in *BRAF$^{V600E}$*-expressing RPE-1 FUCCI cells. Treatment with either inhibitor suppressed the formation of tetraploid cells (Fig. 4C). Furthermore, treatment with either inhibitor led to the reestablishment of equatorial RhoA localization (Fig. 4D, E). Together these data indicate that BRAF$^{V600E}$ acts through Rac1 to inhibit RhoA and cause failure of cytokinesis.

**Supernumerary centrosomes are observed in *BRAF$^{V600E}$*-expressing cells.** In assessing how Rac1 activity could be upregulated in *BRAF$^{V600E}$*-expressing cells, we discovered that mitotic spindles in *BRAF$^{V600E}$*-expressing cells showed structural organizations consistent with the presence of multiple, clustered centrosomes at the same spindle pole. Extra centrosomes increase microtubule nucleation, which in turn stimulates the activity of Rac1[66,67]. To determine whether extra centrosomes were present in *BRAF$^{V600E}$*-expressing cells, we performed quantification of centrosomes via gamma-tubulin staining in S phase cells. Here, we observed a significant increase in cells with more than 2 centrosomes in the *BRAF$^{V600E}$*-induced population (Supplementary Fig. 5A, B). Gamma-tubulin can stain fragments, derived from previously intact centrosomes, that can continue to nucleate microtubules[68]. To address this possible artifact and confirm the presence of supernumerary centrosomal components in *BRAF$^{V600E}$*-expressing cells, we stained for the centriolar marker Centrin-2 in mitotic cells (Fig. 5A). Supernumerary Centrin-2-positive centrioles (>4 centrioles per cell) were observed in 22% of *BRAF$^{V600E}$*-expressing cells, which is an eight-fold increase as compared to control cells (Fig. 5B). In some cases, unpaired, single centrioles, suggestive of centriole overduplication were observed (Supplementary Fig. 5C). To determine if supernumerary centrioles were able to accumulate pericentriolar

material (PCM) and nucleate microtubules, we stained for the PCM marker Pericentrin and alpha-tubulin to visualize microtubules. Cells that had three or more foci of centrioles were assessed so PCM at a supernumerary centriole could be distinguished. We found that 91% of cells had Pericentrin at three or more foci. (Supplementary Fig. 5D). For alpha-tubulin staining, cells that had two or more foci of centrioles at a spindle pole were assessed so microtubules at a supernumerary centriole could be distinguished. We found that 79% of cells had microtubules emanating from two or more foci (Supplementary Fig. 5E). These data indicate that supernumerary centrioles arose as a consequence of BRAFV600E expression and many of these centrioles were competent to accumulate PCM and nucleate microtubules. Supernumerary centrioles were also present in *BRAF$^{V600E}$*-expressing human Mel-ST cultured melanocytes (Supplementary Fig. 6A, B) and *BRAF$^{V600E}$*-expressing zebrafish melanocytes (Fig. 5C, D, Supplementary Fig. 6C, D), supporting the relevance of observations in RPE-1 cells to melanocyte and melanoma biology. To determine whether supernumerary centrioles arose due to BRAF$^{V600E}$-dependent MAPK signaling activity, we treated *BRAF$^{V600E}$*-expressing cells with the MEK inhibitor trametinib and ERK inhibitor SCH772984. Both inhibitors suppressed the increase in centrioles (Fig. 5A, B), indicating that BRAF$^{V600E}$-dependent overactivation of MAPK signaling led to supernumerary centrioles.

**Supernumerary centrioles contribute to RhoA downregulation and *BRAF$^{V600E}$*-induced WGD.** Our findings indicated that BRAF$^{V600E}$ caused RhoA downregulation and failure of cytokinesis as well as an increase in centrioles. We sought to determine whether the increase in centrioles contributed to RhoA downregulation and tetraploid cell formation. First, we examined if *BRAF$^{V600E}$*-expressing cells with extra centrioles had reduced RhoA localization as compared to *BRAF$^{V600E}$*-expressing cells with normal centrioles. *BRAF$^{V600E}$*-expressing cells with a normal number and arrangement of centrioles had reduced RhoA staining; however, *BRAF$^{V600E}$*-expressing cells with extra centrioles had an even greater reduction in RhoA localization (Fig. 6A, B). To establish that the reduced localization of RhoA in these cells was dependent on extra centrioles, we treated cells with centrinone, a PLK4 inhibitor that blocks centrosomal duplication[69]. Centrinone treatment suppressed the formation of extra centrioles in *BRAF$^{V600E}$*-expressing cells (Fig. 6C). The centrinone-treated *BRAF$^{V600E}$*-expressing cells had higher RhoA

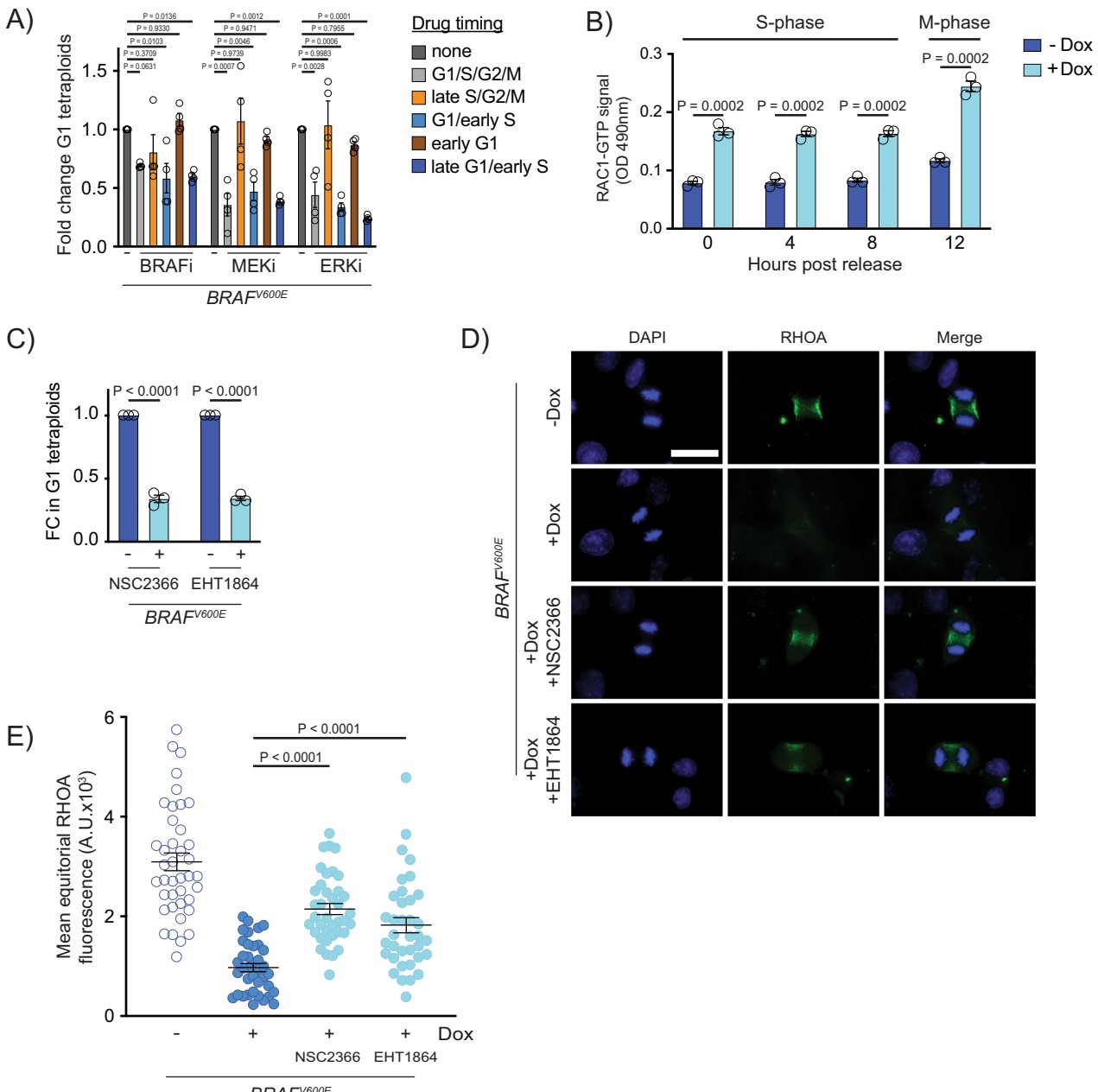

**Fig. 4 BRAF$^{V600E}$ acts during G1/S and activates RAC1 to downregulate RhoA and generate tetraploid cells. A** Fold change in G1 RPE-1 FUCCI tetraploids following inhibitor treatment. Fold changes are expressed relative to control ($+BRAF^{V600E}$, no drug) samples. Inhibitors were added at indicated timepoints. $N = 4$ independent experiments. One-way ANOVA with Dunnett's multiple comparisons test. Error bars represent mean ± SEM. **B** ELISA-based quantification of RAC1-GTP levels in control ($-$Dox) and $BRAF^{V600E}$-expressing ($+$Dox) RPE-1 cells. Cells were measured at the indicated timepoints post thymidine release. RAC-1 GTP signal was measured using a colorimetric assay at 490 nM absorbance. $N = 3$ independent experiments. Unpaired Student's $t$ test. Error bars represent mean ± SEM. **C** Fold change (FC) in G1 RPE-1 FUCCI tetraploids following addition of RAC1 inhibitors. NSC2366 and EHT1864 were added coincident with $BRAF^{V600E}$ induction. Fold changes are expressed relative to control ($+BRAF^{V600E}$, no drug) cells. $N = 3$ independent experiments. Unpaired Student's $t$ test. Error bars represent mean ± SEM. **D** DAPI and anti-RHOA staining in -$BRAF^{V600E}$ ($-$Dox) cells, $BRAF^{V600E}$-expressing ($+$Dox) cells, and $BRAF^{V600E}$-expressing ($+$Dox) RPE-1 cells treated with NSC2366 or EHT1864. Drugs were added coincident with Dox administration. Images are maximum intensity projections of z-stacks (0.20 μM). Scale bar = 7.5 μM. **E** Mean RHOA fluorescence intensity at the equator of control RPE-1 cells, $BRAF^{V600E}$-expressing RPE-1 cells, and $BRAF^{V600E}$-expressing RPE-1 cells treated with NSC2366 or EHT1864. Fluorescence intensities (mean gray values) of the equator were measured by sum intensity projections of z-stacks. $N = 40$ cells for $-$Dox, $N = 38$ for $+$Dox, $N = 38$ for $+$Dox $+$NSC2366, and $N = 36$ for $+$Dox $+$EHT1864. Brown-Forsythe and Welch one-way ANOVA with Dunnett's multiple comparisons test. Error bars represent mean ± SEM.

equatorial localization as compared to control $BRAF^{V600E}$-expressing cells (Fig. 6D, E). This recovery of RhoA localization was evident not only when centrinone treatment was coincident with $BRAF^{V600E}$ expression, but also when centrinone treatment was limited to late G1 and early S. Centrinone treatment also

partially suppressed the $BRAF^{V600E}$-driven formation of tetraploid cells (Fig. 6F). Thus, suppression of supernumerary centrioles in $BRAF^{V600E}$-expressing cells recovered not only RhoA localization but also promoted cytokinesis. Taken together, these data indicate that the reduction of RhoA localization and

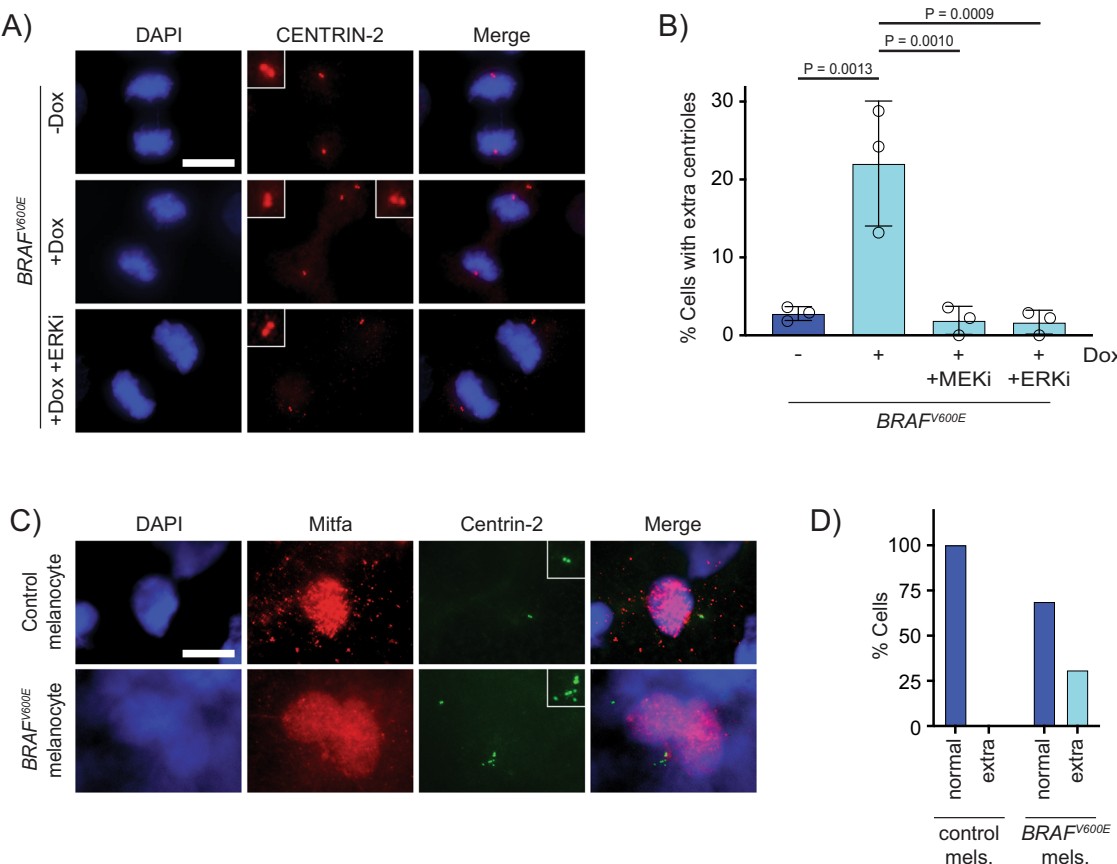

**Fig. 5 Supernumerary centrioles are observed in $BRAF^{V600E}$-expressing cells. A** DAPI and anti-CENTRIN-2 staining in control (−Dox) and $BRAF^{V600E}$-expressing (+Dox) anaphase RPE-1 cells. Insets show centrioles at one pole. Images are maximum intensity projections of z-stacks. Scale bar = 7.5 μM. **B** Percent cells with supernumerary (>4) centrioles in control RPE-1 cells, $BRAF^{V600E}$-expressing cells and $BRAF^{V600E}$-expressing cells treated with MEKi or ERKi. Quantification of RPE-1 cells in mitosis with supernumerary (>4) centrioles. Drugs were added coincident with Dox administration. $N = 3$ independent experiments examining −Dox = 131, +Dox = 169, +Dox +MEKi = 85, and +Dox +ERKi = 91 cells. One-way ANOVA with Tukey's multiple comparisons test. Error bars represent mean ± SEM. **C** DAPI, anti-Mitfa and anti-Centrin-2 staining of control *Tg(mitfa:EGFP); alb(lf)* and *Tg(mitfa:EGFP); Tg(mitfa:BRAF$^{V600E}$); alb(lf)* non-cycling zebrafish melanocytes. Scale bar = 7.5 μM. Insets show centrioles. **D** Percent cells with normal (2) and extra (>2) centrioles in control *Tg(mitfa:EGFP); alb(lf)* and *Tg(mitfa:EGFP); Tg(mitfa:BRAF$^{V600E}$); alb(lf)* zebrafish melanocytes. $N = 36$ melanocytes for *Tg(mitfa:EGFP); alb(lf)* and $N = 50$ for *Tg(mitfa:EGFP); Tg(mitfa:BRAF$^{V600E}$); alb(lf)*. Chi-square test $P = 0.000843$.

cytokinesis in $BRAF^{V600E}$-expressing cells is partially dependent on the $BRAF^{V600E}$-driven increase in centrioles. Because the suppression of supernumerary centrioles in $BRAF^{V600E}$-expressing cells did not fully restore RhoA localization and cytokinesis, a separate, centriole-independent mechanism driven by $BRAF^{V600E}$ also reduces RhoA localization and cytokinesis.

**p53 blocks cell cycle progression of $BRAF^{V600E}$-induced tetraploid cells.** Our finding that $BRAF^{V600E}$ can cause failure of cytokinesis and lead to tetraploidy suggests that this may underlie the genome doubling events that are evident in $BRAF^{V600E}$-mutated melanomas. However, for $BRAF^{V600E}$-expressing cells to contribute to tumor formation, they would likely have to overcome the G1 phase arrest that tetraploid cells experience[21,70]. This tetraploid arrest can be triggered by Hippo pathway activation, which itself is activated by supernumerary centrosome-dependent activation of Rac1[46]. Hippo pathway activation, in turn, activates p53, which has been shown to mediate arrest of tetraploid cells[70,71]. Additionally, inactivation of p53 is strongly correlated with WGD in clinical samples, supporting the possibility of a p53-dependent arrest in tumors[16].

To determine if $BRAF^{V600E}$-expressing tetraploid cells were arrested, we first examined *Tg(mitfa:BRAF$^{V600E}$)* as compared to *Tg(mitfa:BRAF$^{V600E}$); p53(lf)* zebrafish melanocytes. Strikingly,

the nuclei of *Tg(mitfa:BRAF$^{V600E}$); p53(lf)* melanocytes were much larger than those of *Tg(mitfa:BRAF$^{V600E}$)* melanocytes (Fig. 7A, Supplementary Fig. 7A). Since DNA content is correlated with nuclear size[40,72], we determined if *Tg(mitfa:BRAF$^{V600E}$); p53(lf)* melanocyte nuclei had a higher DNA content than those of *Tg(mitfa:BRAF$^{V600E}$)* melanocytes. Using DNA densitometry we found that, whereas *Tg(mitfa:BRAF$^{V600E}$)* and *p53(lf)* nuclei were predominantly 2N, *Tg(mitfa:BRAF$^{V600E}$); p53(lf)* nuclei were mostly 4N (Fig. 7B). This suggests that $BRAF^{V600E}$ expression causes an arrest of tetraploid cells that is dependent on p53. We also analyzed DNA content in zebrafish melanocytes by flow cytometry. GFP-positive melanocytes from *Tg(mitfa:EGFP); p53(lf); alb(lf)* animals were largely diploid as compared to melanocytes from *Tg(mitfa:EGFP); Tg(mitfa:BRAF$^{V600E}$); p53(lf); alb(lf)* animals, most of which had a higher than 4N DNA content (Supplementary Fig. 7B), consistent with results from the nuclear DNA content densitometry analysis. These data suggest that, at least one and sometimes additional rounds of DNA replication can occur in the same melanocyte. The ploidy defects are dependent on both $BRAF^{V600E}$ expression and *p53* deficiency, mirroring the genetic synergism displayed by *Tg(mitfa:BRAF$^{V600E}$)* and *p53(lf)* in melanoma formation.

To more directly address a p53-dependent arrest, we used Crispr/Cas9-mediated genome editing to knock out P53 in our

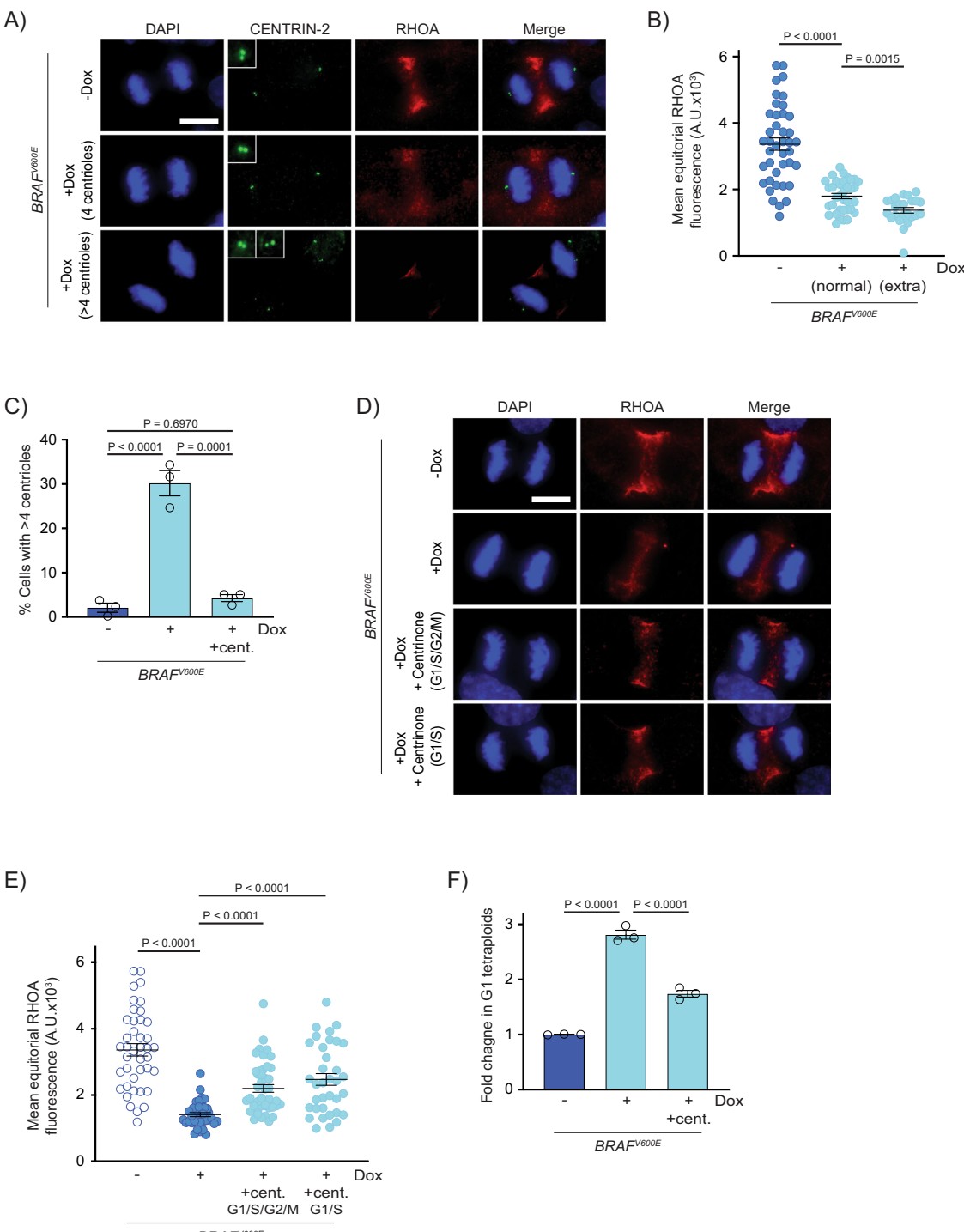

10    NATURE COMMUNICATIONS | (2022)13:4109 | https://doi.org/10.1038/s41467-022-31899-9 | www.nature.com/naturecommunications

*BRAF^{V600E}*-inducible RPE1-FUCCI cell clone. We isolated a clone in which both *P53* alleles were targeted and P53 protein expression abrogated (Supplementary Fig. 7C, D). These cells had DNA content profiles similar to P53 wild-type parental cells (Supplementary Fig. 7E, F). As compared to parental *BRAF^{V600E}*-inducible RPE1-FUCCI cells, *BRAF^{V600E}*-inducible *P53^{−/−}* RPE-1 FUCCI cells had a lower fraction of G1 tetraploids following *BRAF^{V600E}* expression (Supplementary Fig. 7G, H), consistent with the notion that an arrest of G1 tetraploids was bypassed in *P53^{−/−}* cells. Such a failure to arrest would be evidenced by the presence of Geminin-GFP-positive S/G2/M cells with a >4N DNA content. Such cells were present at a much higher fraction

in *BRAF^{V600E}*-inducible *P53^{−/−}* RPE-1 FUCCI cultures and were observed after Dox withdrawal in extended cultures (Supplementary Fig. 7I, J), suggesting that, once generated, tetraploid cells could continue to cycle in a *P53^{−/−}* background. To more directly determine if a P53-dependent arrest prevents *BRAF^{V600E}*-expressing tetraploids from entering the cell cycle, we isolated G1 tetraploids after release from synchronization, cultured them for 24 h, then assessed cell cycle progression. As measured by Geminin-GFP expression, *BRAF^{V600E}*-expressing RPE1-FUCCI cells showed little to no progression (Fig. 7C, D). By contrast, nearly 15% of *BRAF^{V600E}*-expressing p53^{−/−} RPE-1 FUCCI cells were Geminin-GFP-positive. The majority of these Geminin-

**Fig. 6 Supernumerary centrioles contribute to RhoA downregulation and BRAF^V600E-induced tetraploidy. A** DAPI, anti-CENTRIN-2 and anti-RHOA staining in control (−Dox) and *BRAF^V600E*-expressing (+Dox) anaphase RPE-1 cells. Insets show centrioles at one pole. Images are maximum intensity projections of z-stacks. Scale bar = 7.5 µM. **B** Mean RHOA fluorescence intensity at the equator of control RPE-1 anaphase cells (*n* = 30) and *BRAF^V600E*-expressing RPE-1 anaphase cells with normal (4) and supernumerary (>4) centrosomes. *N* = 41 cells for −Dox, *N* = 35 for +Dox with normal centrioles, and *N* = 24 for +Dox with supernumerary centrioles. Brown-Forsythe and Welch one-way ANOVA with Dunnett's multiple comparisons test. Error bars represent mean ± SEM. **C** Percent RPE-1 cells in mitosis with supernumerary (>4) centrioles in control RPE-1 cells, *BRAFV600E*-expressing RPE-1 cells and *BRAFV600E*-expressing RPE-1 cells treated with Centrinone. Centrinone was added coincident with Dox administration. *N* = 3 independent experiments examining −Dox = 119, +Dox = 124, +Dox +Centrinone = 122 cells. One-way ANOVA with Tukey's multiple comparisons test. Error bars represent mean ± SEM. **D** DAPI and anti-RHOA staining in -*BRAF^V600E* (−Dox) RPE-1 cells, *BRAF^V600E*-expressing (+Dox) RPE-1 cells, and *BRAF^V600E*-expressing (+Dox) RPE-1 cells treated with Centrinone. Centrinone was added coincident with Dox administration (G1/S/G2/M) or only during G1/S. Images are maximum intensity projections of z-stacks (0.20 µM). Scale bar = 7.5 µM. **E** Mean RHOA fluorescence intensity at the equator of control RPE-1 cells, *BRAF^V600E*-expressing RPE-1 cells, and *BRAF^V600E*-expressing RPE-1 cells treated with Centrinone coincident with Dox administration (G1/S/G2/M) or only during G1/S. *N* = 41 cells for −Dox, *N* = 39 for +Dox, *N* = 44 for +Dox +Centrinone in G1/S/G2/M, and *N* = 35 for +Dox +Centrinone in G1/S. Brown-Forsythe and Welch one-way ANOVA with Dunnett's multiple comparisons test. Error bars represent mean ± SEM. **F** Fold change in G1 tetraploids in control RPE-1 FUCCI cells (−Dox), *BRAF^V600E*-expressing RPE-1 FUCCI cells (+Dox) and *BRAF^V600E*-expressing RPE-1 FUCCI cells treated with Centrinone (+Dox +cent.). Fold changes are expressed relative to the control cells. *N* = 3 independent experiments. One-way ANOVA with Tukey's multiple comparisons test. Error bars represent mean ± SEM.

GFP-positive cells had DNA content reflective of progression into S/G2/M cell cycle phases (Fig. 7C). Together these data indicate that *BRAF^V600E*-induced tetraploid cells arrest in G1, and this arrest is alleviated by loss of P53. Our analysis of human tumor samples confirmed that loss of p53 pathway activity was strongly correlated with WGD, including in tumors that harbor BRAF mutations and other mutations that activate RAS/MAPK signaling (Supplementary Fig. 8).

**Nascent tumor cells in *BRAF^V600E*-driven zebrafish melanomas are tetraploid and have higher ploidy**. Our data indicate that BRAF^V600E can induce tetraploidy in melanocytes, and these melanocytes are prevented from progressing further by a p53-dependent block. Since *BRAF^V600E* and loss of *p53* cooperate to form melanomas, and because *BRAF^V600E* causes tetraploidy as seen in our in vivo and in vitro models, we were interested in understanding whether *BRAF^V600E*-generated tetraploid cells serve as intermediates in melanomagenesis in our zebrafish model, potentially as cells of origin. This would be consistent with findings in tumor types with frequent WGD, in which tumors that have undergone a WGD event are thought to have undergone that event early in tumor progression[3,73]. To determine if tetraploid cells are present in the earliest stages of tumor formation, we took advantage of the zebrafish model in which early melanomas can be identified because of their reactivation of the neural crest gene *crestin*[74]. Using a *Tg(mitfa:BRAF^V600E); p53(lf); Tg(crestin:EGFP)* strain that marks tumor initiating cells, we identified zebrafish with early tumors (<20 cells), dissected these tumors and performed flow cytometry to determine DNA content of *crestin:EGFP*-positive tumor cells. The *crestin:EGFP*-positive cells were mostly tetraploid, and several octoploid cells were also observed (Fig. 7E). Notably, none of the cells were diploid, indicating that cells were tetraploid very early in tumor formation and it is likely that the cell of origin was tetraploid. Furthermore, the octoploid nascent tumor cells were mostly mononucleate (Fig. 7E) and thus different from the octoploid, binucleate melanocytes present in *Tg(mitfa:BRAF^V600E); p53(lf)* strains. We speculate that these mononuclear, octoploid tumor cells arose from tetraploid cells of origin, and were cycling tetraploid cells in G2 and M phases. These data suggest that WGD can be present at the time of tumor initiation and potentially support tumor progression.

## Discussion

Our results indicate that BRAF^V600E, and RAS/MAPK signaling in general, can cause the WGD that is a hallmark of melanomas and other tumor types. BRAF^V600E causes WGD by suppressing the activity of RhoA, leading to failure of cytokinesis. This suppression stems, in part, from supernumerary centrosomes that are formed as a result of BRAF^V600E activity. Supernumerary centrosomes are known to activate Rac1, and we found that Rac1 activity was necessary for BRAF^V600E-dependent WGD. Taken together, our data support a model in which BRAF^V600E causes the formation of supernumerary centrioles and centrosomes, leading to the activation of Rac1, which in turn causes inhibition of RhoA and failure of cytokinesis. The binucleate, tetraploid cells that result arrest in G0/G1 unless cells have an underlying mutation, in our models it is loss of P53, that abrogates the arrest and enables further cell cycle progression. Thus, in addition to stimulating cell cycle progression, suppressing cell death and providing other tumor-promoting activities, BRAF^V600E can cause WGD, which has been shown to support tumorigenesis and tumor progression.

A key component of this model is the formation of supernumerary centrosomes in *BRAF^V600E*-expressing cells. The requirement for BRAF^V600E activity in G1/S coincides with the timing of centrosomal duplication in most cell types. Furthermore, gamma-tubulin staining in *BRAF^V600E*-expressing cells showed an increase in centrosome numbers in cells that had only progressed into S and G2 phases after *BRAF^V600E* induction in G1. Together these observations suggest that the defect caused by BRAF^V600E is that of centrosomal overduplication. A similar phenotype has been observed upon *BRAF^V600E* overexpression in established melanoma cell lines, although such cells had a high background of underlying centrosomal abnormalities[75,76]. More recently, direct staining of melanoma samples suggests that supernumerary centrosomes in patient samples arise predominantly through overduplication[77]. The mechanism by which BRAF^V600E could cause overduplication is not clear, although the ability of centrinone to suppress *BRAF^V600E*-induced supernumerary centrioles indicates that this mechanism is PLK4-dependent. Our data also indicate that supernumerary centrioles accumulate pericentriolar material (PCM) and have microtubule organizing center (MTOC) activity once they are formed. This is surprising because, under normal circumstances, newly-formed centrioles need to traverse through mitosis to accumulate PCM and nucleate microtubules[78]. How precocious maturation of *BRAF^V600E*-induced supernumerary centrioles could occur is not entirely clear, although a combination of studies suggest premature centriole disengagement might be important. RAS/MAPK pathway activity, such as that induced by BRAFV600E, is a well-established cause of replication stress[79,80]. Recently, it has been

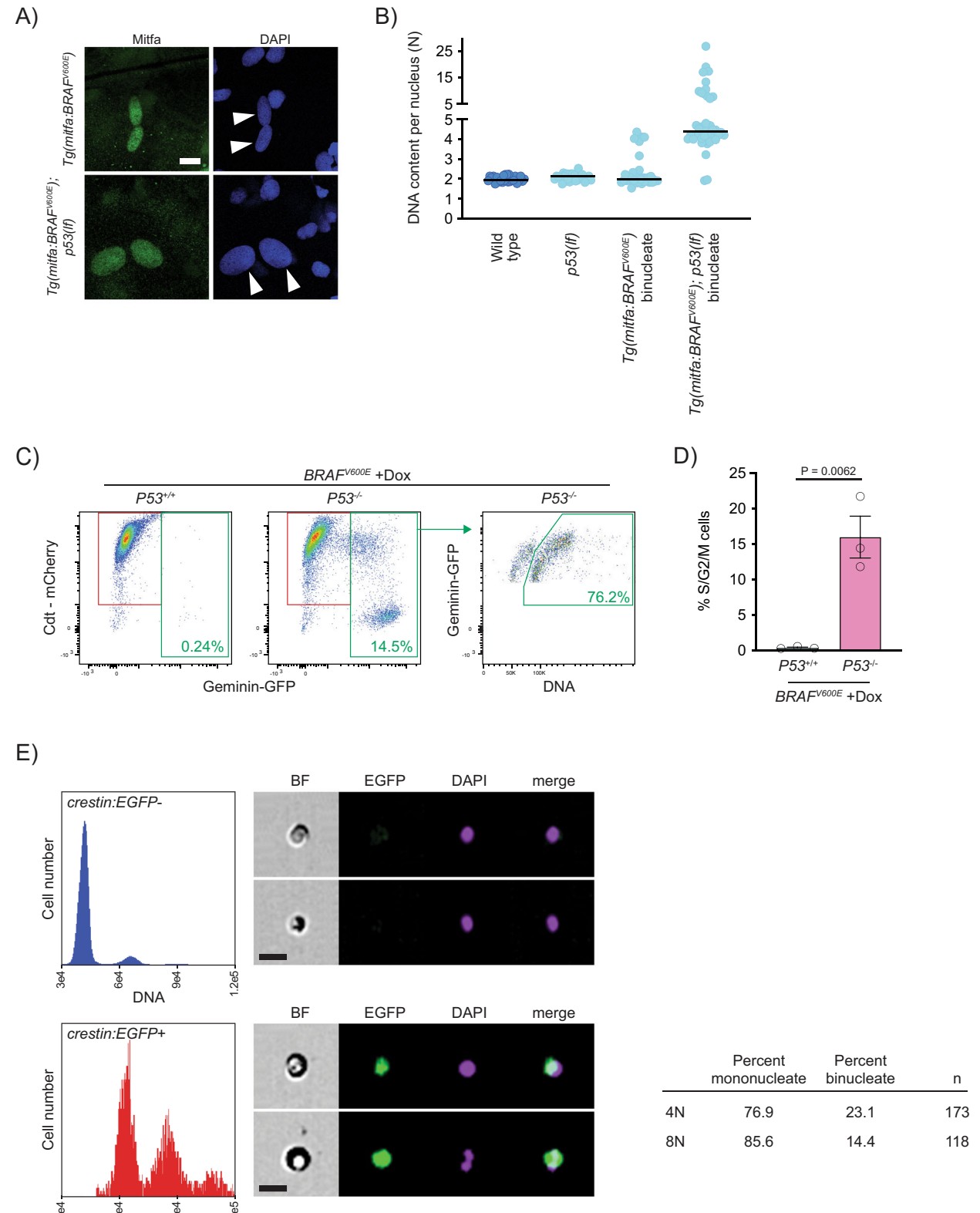

shown that replication stress causes premature centriole disengagement, leading to the formation of multipolar spindles that show microtubule nucleation on prematurely disengaged centrioles[81]. Premature centriole disengagement caused by disruption of *Cep57* and PCM components also leads to precocious maturation, suggesting that premature disengagement can generally enable precocious maturation and MTOC activity[82]. Thus,

taken together, RAS/MAPK-induced replication stress caused by *BRAF^{V600E}* expression could lead to premature centriole disengagement followed by precocious maturation and MTOC activity. As noted, the effect of supernumerary centrosomes only accounts for some of the RhoA reduction and cytokinesis failure upon *BRAF^{V600E}* induction. Nonetheless, our findings together with the association of BRAF^{V600E} with centrosomal amplification in

**Fig. 7 Progression of p53-mutant BRAF$^{V600E}$-induced tetraploid cells. A** Anti-Mitfa and DAPI staining of nuclei in *Tg(mitfa:BRAF$^{V600E}$)* and *Tg(mitfa:BRAF$^{V600E}$); p53(lf)* melanocytes. Melanin pigment was bleached to more clearly visualize nuclear size. White arrowheads indicate two nuclei within a single melanocyte. Scale bar = 10 μm. **B** DNA content per nucleus as measured by confocal densitometry. Each data point represents a single nucleus. N = 25 nuclei for wild type, N = 31 for *p53(lf)*, N = 35 for binucleate *Tg(mitfa:BRAF$^{V600E}$)*, and N = 42 for binucleate *Tg(mitfa:BRAF$^{V600E}$);p53(lf)* cells. Bars represent median. **C** Flow cytometry plots of *BRAF$^{V600E}$*-expressing (+Dox) p53 wild-type (left) and p53 mutant (middle) tetraploid cells. Prior to analysis, RPE-1 FUCCI G1 tetraploid cells were isolated and plated for 24 h in the presence of nocodazole. Percentages of Geminin-GFP-positive cells in S/G2/M are indicated. Flow cytometry plot of *BRAF$^{V600E}$*-expressing (+Dox) p53 mutant S/G2/M cells (right). Percentage of ≥4 N S/G2/M cells is indicated. **D** Percent S/G2/M cells from flow cytometry analysis in (**C**). N = 3 independent experiments. Unpaired Student's *t* test. Error bars represent mean ± SEM. **E** Flow cytometry and DNA content analysis of normal tissue (left, top) and nascent melanomas (left, bottom) from *Tg(crestin:EGFP); Tg(mitfa:BRAF$^{V600E}$); p53(lf)* zebrafish. Brightfield, EGFP and DAPI images of single melanocytes are shown. Quantification of percent mononucleate and binucleate cells in the 4N and 8N peaks of nascent melanomas (right).

papillary thyroid, colorectal and other cancers suggests there may be a broad link between BRAF$^{V600E}$-induced supernumerary centrosomes and WGD[83,84].

Our findings begin to address the fate of cells that undergo WGD through cytokinesis failure. Zebrafish melanocytes and RPE-1 cells that have undergone cytokinesis failure arrest in G1 as binucleate tetraploid cells. However, in established zebrafish melanomas and late-stage human tumors that have previously experienced a WGD event, tumor cells go through the cell cycle, are predominantly mononuclear and do not indefinitely undergo cytokinesis failure. This raises the questions of how cells having just undergone a WGD event overcome the G1 block and, once they do, divide productively without creating giant, multinucleated cells. Newly-generated G1 binucleate tetraploid cells undergo a P53-dependent arrest, as evidenced by the progression into S phase of P53-mutant RPE-1 G1 tetraploids. The polyploid nature of *Tg(mitfa:BRAF$^{V600E}$); p53(lf)* melanocytes indicates that this arrest also occurs in the zebrafish model. However, these zebrafish melanocytes are found as binucleates with most nuclei appearing to have gone through a single S phase without any further mitosis. This suggests that additional blocks may exist to prevent such cells from progressing further into the cell cycle, and such blocks would have to be overcome during tumorigenesis. If such cells were to progress and continue cycling, then conditions would have to exist to prevent them from stalling tumor growth as multinucleated cells. One possible condition is that the penetrance of BRAF$^{V600E}$-induced cytokinesis failure is incomplete as in RPE-1 cells. Enough cytokinesis failure to enable WGD would be present, but not enough to prevent the amount of cell divisions required for tumor growth. Another possible condition is the attenuation of cytokinesis failure following a WGD event. RhoA activity has been shown to be upregulated in tumors[85–87], and this could be an adaptation that prevents pervasive cytokinesis failure that would be detrimental to tumor progression.

Our finding that nascent zebrafish melanoma cells were tetraploid or had higher ploidy supports the notion that BRAF$^{V600E}$-induced WGD can be present at the time of tumor initiation. In cancer types that have a high prevalence of WGD, such as melanoma, WGD in tumors frequently occurred early in tumor formation and might have been present at the time of tumor initiation and clonal outgrowth[3,73]. This raises the question of whether BRAF$^{V600E}$, which is present in ~80% of benign nevi[88,89], can cause WGD at the earliest stages of tumor formation and possibly in benign lesions. Although cells in common cutaneous nevi are primarily diploid[90,91], several observations suggest that cytokinesis failure and associated WGD can occur in these cells. Cultures of nevocytes have been shown to contain binucleated cells[92]. Furthermore, the presence of binucleate and multinucleated giant cells in nevi is not uncommon[93], and polyploid cells are observed at a low fraction in many nevus samples[94]. Together these observations suggest that some level of cytokinesis failure occurs in nevi and may be caused by

BRAF$^{V600E}$. The prevalence of early WGD in melanomas as compared to its relative absence in nevi, at least its absence when nevi begin clonal outgrowth, suggests there is some advantage to WGD in tumor formation. The ability to sample various pro-tumorigenic genomic configurations may underlie the benefit of WGD[95] as would the ability of WGD to make haploid regions of a nascent tumor cell diploid and thus protect these regions from mutations incurred in essential genes[24]. Lastly, although the focus has been on WGD supporting tumor initiation, it is clear that many tumors analyzed bioinformatically, including the majority (~60%) of melanomas, show no evidence of having undergone WGD. However, this does not mean WGD is irrelevant in such tumors. Recent evidence suggests that WGD in melanomas can be late truncal and even occur privately in metastases[96,97]. Therefore, while WGD may not be necessary in all tumors for initiation, it nonetheless could play a role in disease progression. It is worth noting that the WGD observed in nevocytes, primary tumors and metastases overlaps with the presence of BRAF$^{V600E}$ or other RAS/MAPK-activating mutations that could support WGD in such lesions.

## Methods

**Zebrafish strains and husbandry**. Zebrafish were handled in accordance with protocols approved by the University of Massachusetts Medical School IACUC (A-2171). Strains were maintained at 28 °C as described by Westerfield[98]. AB was used as the wild-type strain. The following mutations were used: *p53(zdf1)*[99], *mitfa(w2)*[100], *alb(b4)*[101] and are designated throughout as "*lf*" for loss-of-function mutation. The integrated transgene *Tg(mitfa:BRAF$^{V600E}$)* expresses oncogenic human *BRAF$^{V600E}$* under the control of the zebrafish pigment cell-specific *mitfa* promoter[38]. Equal numbers of male and female animals were used in experiments.

**Cell line generation**. RPE-1 FUCCI cells were grown in DMEM: F12 media containing 10% tetracycline-free FBS, 100 IU/ml penicillin, and 100 μg/ml streptomycin. Mel-ST cells were grown in DMEM media containing 5% tetracycline-free FBS, 100 IU/ml penicillin, and 100 μg/ml streptomycin. Cells were maintained at 37 °C with 5% CO$_2$ atmosphere. TetR was expressed in cells using pLenti CMV TetR Blast (Addgene #17492). Transduced cells were selected with 5 μg/ml blasticidin and bulk cells expressing the TetR protein were clonally isolated in 96-well plates using FACS (BD-Aria). *BRAF$^{V600E}$* and *BRAF$^{WT}$* were expressed in cells using Gateway-compatible lentiviral constructs (pLenti CMV/TO puro DEST, pLenti CMV/TO neo DEST; gifts from Eric Campeau and Paul Kaufman[102]). Lentivirus carrying *BRAF$^{WT}$* or *BRAF$^{V600E}$* construct was generated by transfection of HEK-293T cells, with appropriate packaging plasmids (pMD2.G and psPAX2) using lipofectamine 2000, according to the manufacturer's instructions. RPE-1 and Mel-ST cells were infected for 48 h with virus carrying a BRAF construct in the presence of 8 μg/ml polybrene, washed, and allowed to recover for 24 h before selection. Cells were selected with the appropriate antibiotic selection marker (5 μg/ml puromycin or 1 mg/ml neomycin) and clonally isolated in 96-well plates using FACS (BD-Aria). Doxycycline was used at 1 μg/ml to induce the expression of *BRAF*.

**Synchronization and ploidy analysis**. To turn on *BRAF$^{V600E}$* in a synchronized cell population, we serum starved RPE-1 FUCCI cells with 0.1% serum for 48 h. Serum-starved cells were released into media with 10% serum until they reached the next G1 (~33 h). *BRAF$^{V600E}$* was turned on using doxycycline, and cells were resynchronized at G1/S using thymidine (2.5 mM). Following thymidine washout cells were allowed to progress through the cell cycle, and ploidy was analyzed in next G1 (16 h after thymidine release) using Hoechst incorporation (2.5 μg/ml) in

hCdt1-mcherry expressing cells by flow cytometry, which was performed using a BD-LSR flow cytometer, acquired using BD FACS Diva software (v. 8.0.1) and analyzed using FlowJo software (v. 10.0). For immunofluorescence experiments of mitotic cells, cells were fixed at 12 h after thymidine release.

**Live cell imaging**. For live cell imaging, RPE-1 H2B-GFP-transfected cells were grown on glass-bottom 12-well tissue culture dishes (MatTek) and imaged on a Nikon TE2000E inverted microscope equipped with a cooled Hamamatsu Orca-ER CCD 11 camera and the Nikon Perfect Focus system. The microscope was enclosed within an incubation chamber that maintained an atmosphere of 37 °C and 3–5% humidified $CO_2$. GFP and phase-contrast images were captured at multiple locations every 5 min with either 20× or 40× Nikon Plan Fluor objectives. All captured images were analyzed using NIS-Elements software. For imaging, cells were synchronized with thymidine and imaged for 48 h after release. Each cell going through mitosis was observed and cytokinesis was scored by phase contrast imaging. To verify cytokinesis failure and to confirm that the nuclei stayed within one cell, cells were carefully traced for at least ten frames. To quantify mitotic duration, each frame beginning from the start of nuclear envelope breakdown to anaphase onset was counted.

**Immunofluorescence and imaging**. RPE-1 BRAF^V600E cells were grown directly on collagen IV (Sigma) -coated coverslips, fixed in 3.7% formalin, permeabilized using 0.1% Triton X-100, and treated with 0.1% SDS. They were blocked in 1% BSA and then incubated with primary antibody diluted in blocking solution in a humidity chamber at 4 °C overnight, washed with 1X PBS, and incubated with secondary antibody. Cells were mounted using mounting media containing DAPI (Vector Laboratories). All secondary antibodies were Alexa Fluor conjugates (488, 555, and 647) (Thermofisher) used at a 1:500 dilution. All images were acquired on a Leica DM 600 inverted microscope at 100× magnification and analyzed using Leica LAS X software (v. 3.7.1.21655). Z stack images were taken at 0.20 μM each. For melanocyte binucleate counts and immunofluorescence staining, zebrafish adult dorsal scales were fixed using 4% paraformaldehyde for 2 h, washed with PBST (PBS + 0.1% Triton X) and water then blocked in 1%BSA/PBS for 30 min prior to primary antibody incubation. Affinity-purified anti-Mitfa antibodies were used at a 1:100 dilution for staining. All images were acquired on a Nikon Eclipse Ti A1R-A1 confocal microscope and analyzed using NIS-Elements software.

**RhoA and Rac1 activation assays**. RPE-FUCCI cells were synchronized as described above, and lysates were collected at 0, 4, 8, 12 h post release from thymidine. GTP-bound RhoA and Rac1 were immunoprecipitated from cells according to manufacturers' protocols (BK-036, BK-128, Cytoskeleton).

**Western blotting and antibodies**. Cells were lysed in ice-cold RIPA buffer containing a Complete protease inhibitor tablet (Roche). Protein concentration was measured using the Pierce BCA Protein Assay Kit (Life Technologies). Samples were run on 10% polyacrylamide gels, transferred, and developed using fluorophore-conjugated antibodies (LI-COR). Primary antibodies and dilutions used were: Total BRAF, 1:1000 (10146, Millipore); TetR, 1:1000 (631132, Takara); TUBULIN 1:1000 (2144S, Cell Signaling); BRAF^V600E 1:1000 (E19290, Spring Bioscience); RHOA(GTP) 1:500 (ARH03, Cytoskeleton); Total RHOA, 1:500 (ARH05, Cytoskeleton); RAC1 1:500 (ARC03, Cytoskeleton); p-MPS1 T676 1:1000 (PA5-64614, Thermofisher); Total MPS1 (PA5-116969, Thermofisher); p-MEK 1:1000 (9154, Cell Signaling); Total MEK 1:1000 (8727, Cell Signaling); p-ERK 1:1000 (m8159, Sigma); Total ERK 1:1000 (4695, Cell Signaling); P53 1:1000 (sc-126, SCBT); GAPDH 1:5000 (AM4300, Thermofisher). Secondary antibodies and dilutions used were: IRDye 800CW Donkey anti-Rabbit 1:10,000 (LI-COR 926-32213); IRDye 680RD Goat anti-Mouse 1:10,000 (LI-COR 926-68070). Uncropped and unprocessed scans of western blots are provided in Source Data.

**Cyclin D1 staining for flow cytometry**. RPE-1 FUCCI cells were trypsinized, washed with 1XPBS and fixed in 2% PFA at RT for 15 min. Cells were then washed with PBS/1%FBS and permeabilized with 100% methanol (for CYCLIN D1). Cells were washed with PBS, incubated with Alexa Fluor 647-conjugated primary antibodies for 60 min at room temperature, washed and stained with 1:1000 DAPI for 15 min before flow cytometry (BD-LSR).

**CRISPR cell line generation**. Oligos targeting p53 were annealed and inserted into lentiCRISPR v2 (gift of Feng Zhang) as described previously[103,104] The lentiviral packaging plasmids pMD2.G (Addgene plasmid #12259) and psPAX2 (Addgene plasmid #12260) were used for transfection using lipofectamine. For lentiviral transduction, cells (300,000) were plated in a 6 well plate the day before transduction. Lentivirus was harvested and added to OptiMEM supplemented with 8 μg/mL polybrene. Media was changed 24 h after transduction to remove polybrene. Media supplemented with 5 μg/mL puromycin (Sigma Aldrich) or was changed 48 h after transduction to select lentiCRISPRv2 transduced cells. Following transduction with p53-targeting lentiCRISPRv2 containing lentivirus, antibiotic-resistant cells were selected then clonally isolated by FACS in 96-well plates. Clones were then assayed for indels via the surveyor assay (706020, IDT). Cells that

showed indels were then cloned into the pGEM-T Easy vector (Promega), and colonies were picked using blue/white screening then sequenced. The p53 gRNA sequence was CCCCGGACGATATTGAACAA. Primers used for sequencing indels were as follows:

Forward – 5′ GTAAGGACAAGGGTTGGGCT 3′
Reverse – 5′ GAAGTCTCATGGAAGCCAGC 3′

**Flow cytometry of zebrafish melanocytes, normal tissue and tumors**. To isolate zebrafish melanocytes for flow cytometry, 4- to 6-month-old fish were treated for 5 min with the anesthetic tricaine methanesulfonate. Scales were plucked from the dorsal anterior region of fish and put in 0.9× PBS. Cells were dissociated using TH liberase (Roche) for 30 min by constant agitation at 37 °C. Cells were immediately fixed for 2 h in 4% paraformaldehyde. Following fixation, cells were washed and permeabilized 3× with 0.1% Triton X/PBS and immediately stained with DAPI (1:1000). Cells were then filtered through a 40 μM mesh filter and spun down at 2000rpm. Analysis for GFP+ and DAPI+ cells was performed on the Amnis Flowsight using IDEAS software (v. 6.0). For ploidy analysis of zebrafish tumors, melanomas from *Tg(mitfa:EGFP); (mitfa:BRAF^V600E); p53(lf);alb* animals were removed, homogenized in PBS, and stained and analyzed in the same manner as described above. For normal tissue analysis, the caudal peduncle and fin were dissected and homogenized.

**Confocal densitometry**. Zebrafish scales were obtained and stained with anti-Mitfa antibody and DAPI as described above. Melanin pigment interferes with quantitative UV-based imaging so scales were bleached prior to staining. Mitfa-positive melanocyte nuclei were identified, and Z stacks of the DAPI signal of these nuclei were obtained. The same distance between Z slices (0.37 μM) and pixel intensity lookup table were used for each nucleus measured. In analyzing each slice, nuclear boundaries were specified, and pixel intensity values within the nuclear area measured. Pixel intensities for all slices of one nucleus were summed to derive a raw DNA content. Ploidy was estimated using nearby Mitfa-negative nuclei as 2 N controls. Nuclei in the same binucleate cell are typically arranged as mirror-image pairs, and this orientation allowed us to identify nuclei of binucleate cells in the absence of melanin pigment.

**Melanocyte density assay**. Four- to six-month-old fish were treated for 5 min with the anesthetic tricaine methanesulfonate and epinephrine, which contracts melanosomes to the central cell body of melanocytes (Goodrich and Nichols, 1931), thereby resolving overlapping cells. Scales were plucked from the dorsal anterior region of fish from the scale rows adjacent to the dorsal midline row. Scales were immediately fixed for ≥30 min in 4% paraformaldehyde. After fixation, scales were flat-mounted and melanocytes counted. Area was estimated by multiplying maximal antero-posterior and left-right distances of the scale.

**Data collection and statistical analysis**. Microsoft Excel (v. 16.16.27) and Graphpad Prism (v. 8.0) were used for data collection. Significance calculations were performed on samples collected in a minimum of biological triplicate. $P$ values from two-tailed Student's $t$ tests or ANOVA were calculated for all comparisons of continuous variables. All further significance tests were performed in GraphPad Prism (v. 8.0). A $P$ value < 0.05 was considered significant.

**Drug treatments**. Cells were synchronized as described with serum starvation and thymidine arrest. Treatment of cells with small molecule compounds was performed as follows: G1/S/G2/M—drug was added 33 h following serum addition and coincident with doxycycline; S/G2/M—drug was added following thymidine washout; G1/S—drug was added 39 h following serum addition and washed off 12 h later; G1—drug was added 33 h following serum addition and washed off 6 h later; early G1 – drug was added 33 h following serum addition and washed off 2 h later. Small molecule inhibitors used were—Vemurafenib, 1 μM (S1267, Selleck Chemicals), Trametinib 50 nM (S2673, Selleck Chemicals), SCH772984 50 nM (S1701, Selleck Chemicals), PLX7904 1 μM (S7964, Selleck Chemicals), PLX8394 1 μM (HY-18972, MCE), Centrinone 300 nM (5687, Tocris), NSC2366 5 μM (13196, Cayman Chem), EHT1864 5 μM (17258, Cayman Chem), LPA 1 μM (BML-LP100-0005, Enzo), S1P 1 μM (62570, Cayman Chem), Dihydrocytochalasin B 4 μM (D1641, Sigma).

**Immunofluorescence, flow cytometry and antibodies**. For RHOA staining, cells were fixed in 10% Trichloroacetic acid (TCA) (T6399, Sigma) on ice for 15 min, washed with PBS/30 nM glycine, then permeabilized with 0.2% Triton X/PBS/30 nM glycine for 10 min on a rocking platform. Cells were then washed with PBS/30 nM glycine, blocked with 3% BSA/PBS/0.01% Triton X-100 for 1 h at room temperature before staining with primary antibody in a humidity chamber at 4 °C overnight. For ANILLIN Staining, cells were fixed in ice cold methanol at −20 °C for 2 h. Cells were then washed with PBST (PBS/0.01% Triton X-100), permeabilized with 0.2% Triton X-100 in PBS for 10 min, washed with PBST and then incubated with primary antibody in a humidity chamber at 4 °C overnight. To quantify RHOA and ANILLIN staining, fluorescence intensities (mean gray values) of the equator were measured by sum intensity projections of z-stacks. For

CENTRIN-2 staining alone or other markers, cells were fixed in ice cold 100% methanol at −20 °C, permeabilized with 0.2% TritonX/PBS and blocked in 1%BSA/PBS before incubating with primary antibody (1:100) in a humidity chamber at 4 °C, overnight. Primary antibodies and dilutions used for immunofluorescence were: ANILLIN 1:100 (PA5-28645, Thermofisher); RHOA 1:75 (sc-418, SCBT); CENTRIN-2 1:100 (04-1624, Millipore); Mitfa[37] 1:100 (T6557, Sigma); α-TUBULIN DM1A 1:2000 (3873S, Cell Signaling); PERICENTRIN 1:100 (ab4448, Abcam). Antibodies and dilutions used for flow cytometry were: CYCLIN D1, 1:100 (NBP2-33138AF647, Novus Bio), Rabbit IgG Isotype control, 1:100 (3452S, Cell Signaling), HA tag, 1:100 (IC6875R, R&D systems), Mouse IgG1 Isotype control 1:100 (IC002R, R&D systems). Secondary antibodies and dilutions used for immunofluorescence and flow cytometry were: Alexa Fluor 555 goat anti-rabbit IgG 1:500 (A-21429, ThermoFisher); Alexa Fluor 488 goat anti-mouse IgG2a 1:500 (A-21137, ThermoFisher); Alexa Fluor 555 goat anti-mouse IgG1 1:500 (A-21121, ThermoFisher); Alexa Fluor 488 donkey anti-mouse IgG 1:500 (A-21202, ThermoFisher).

**RhoA and Anillin intensity measurements**. Fluorescence intensities were quantified using ImageJ as previously described[105]. Briefly, Z-stack images were loaded on ImageJ (v. 1.53a) and the sum intensity projection was obtained using the Z-stack function. RhoA and Anillin equatorial fluorescence intensities were obtained by measuring the intensity profile of the fluorescence signal along a line manually placed along the cell equator, parallel to the anaphase DNA position. The mean fluorescence intensity was obtained by averaging the two intensity values at each side of the furrow. The mean background signal was obtained by averaging the signal of three manually selected circular regions with a diameter of 50 pixels outside of the cell and the value was subtracted from the equatorial intensities.

**Growth and apoptosis assays**. To perform growth curves, 20,000 cells were plated in 12-well plates and appropriate samples were treated with inhibitors and doxycycline. Cells were counted every 24 h over 4 days using the hemocytometer. Cell death following inhibitor treatment was assessed by the caspase glow assay according to the manufacturer's instructions (G8090, Promega).

**Whole-genome doubling analysis**. Whole-genome doubling (WGD) data for each cohort were obtained for TCGA samples[106] and MSK-IMPACT samples[16], and mutation data were obtained from cBioPortal[107,108]. Associations of WGD with BRAF mutations as well as ANY RAS/MAPK pathway genes (BRAF, NRAS, KRAS, HRAS, NF1, MAP2K1, MAP3K13 and PTPN11) were analyzed. Associations of WGD with BRAF or ANY RAS/MAPK mutations together with TP53 or CDKN2A mutations and copy number losses were also analyzed, with the rationale that our data support a combination of RAS/MAPK activation plus TP53 pathway inactivation enable progression of cells that undergo WGD.

**Reporting summary**. Further information on research design is available in the Nature Research Reporting Summary linked to this article.

## Data availability
All data generated or analyzed in this study are included in this published article and its Supplementary Information file. Source data are provided with this paper.

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

## Acknowledgements

We thank Kristyna Kotynkova for advice on RhoA and Anillin immunofluorescence staining; Paul Kaufman and Eric Campeau for the pLenti-CMV/TO neo/puro plasmids; Daryl Bosco for use of the Leica-DM inverted microscope; Desiree Baron for imaging and microscopy advice; Charles Kaufman for the *Tg(crestin:EGFP)* zebrafish strain; Melissa Guerin for help with fish genotyping; Patrick White, Ed Jaskolski and the staff at the UMMS Animal Medicine Department for fish care; Tammy Krumpoch and Susanne Pechhold at the UMMS Flow Cytometry Core for guidance and assistance in performing flow cytometry and FACS experiments. M.A.V. was supported by 1F30CA228388 and 5T32GM008541. N.J.G. was supported by GM117150, the Harry J. Lloyd Charitable Trust, and the Jackie King Young Investigator Award from the Melanoma Research Alliance. This research was supported by NIAMS of the National Institutes of Health under award number AR063850 and by the USA Department of Defense under award W81XWH2010288 to C.J.C.

## Author contributions

R.D. and C.J.C. conceptualized the study, designed the in vitro experiments, and wrote the manuscript. R.D. and C.J.C. designed all the zebrafish experiments. R.D. and C.J.C. performed all the in vitro biological assays, immunofluorescence staining, flow cytometry and image analysis. M.A.V. and N.J.G. performed the live-cell imaging experiments and R.D. analyzed the data. N.J.G. provided critical reagents.

## Competing interests

The authors declare no competing interests.
