## [Peer Review File · Nature Communications]

Oncogenic BRAF induces whole-genome doubling through suppression of cytokinesisEditorial Note: Parts of this Peer Review File have been redacted as indicated to remove third-party material where no permission to publish could be obtained.

REVIEWER COMMENTS

Reviewer #1 (Remarks to the Author):

In "Oncogenic BRAF Induces Whole-Genome Doubling Through Suppression of Cytokinesis" by Darp et al, the authors use in vivo zebrafish melanoma models and tractable in vitro cell line models to determine the role of overactive BRAF in disrupting the normal processes of cell division/RhoA function and contributing to the abnormal growth/behavior of melanoma cancer. The study addresses interesting questions that touch on important fundamental aspects of cancer biology related to the precise mechanisms of how specific oncogenic events fan out to disrupt normal cell biology and has specific relevance to the relationship between preneoplastic melanocytes, nevi, and frank melanomas. The experiments are overall well done and generally well explained with appropriate conclusions drawn, which will likely be of interest to a wide audience. With a few important weaknesses addressed (below), the manuscript would be appropriate for publication:

Major points:

1) For this reviewer, the use of model organisms (zebrafish) and cell lines (RPE-1 and related) is reasonable and appropriate. However, the authors should better describe the RPE-1 line (presumably TERT immortalized, as in Shenk et al, *Method Mol Biol*, 2016?), and the relationship of this cell type to normal melanoma formation. RPE is a pigmented epithelium but not the precursor to cutaneous melanoma – which is okay for the elegant and precise cell cycle, etc. analysis which is made feasible with the RPE-1 FUCCI and related engineered lines – but this should be better highlighted and more openly explained in results/discussion. Related - might TERT-mediated immortalization of RPE-1 cells have any effect on the underlying cell biological/cytokinesis mechanisms at play?

2) The authors note that supernumerary centrosomes have been noted in established melanoma cell lines and in patient samples, supporting the "in human disease" relevance of the findings. However, without repeating the elegant cell cycle analyses made possible with the RPE-1 FUCCI line and related engineered lines in a melanocyte, the paper would seem to still implicitly argue that BRAF-activation would lead to RhoA mislocalization and accumulation of centrioles soon/immediately after induction of BRAFV600E in a melanocyte system. Such an experiment, which should be feasible with RhoA and Centrin staining in BRAFV600E expressing melanocytes, would strengthen/bridge the RPE-1 cell line work and "melanoma tumor initiation from a melanocyte" context which the paper seems targeted towards.

Minor points:

- Should cite Santoriello et al., "Kita Driven Expression of Oncogenic HRAS Leads to Early Onset and Highly Penetrant Melanoma in Zebrafish" in *PLOS One* (2010) as having seen binucleate melanocyte in this different melanoma model – seems to support the even more broad applicability of these proposed mechanisms to another oncogenic driver. Also, while almost certainly a different phenomenon, worth further contextualizing the paper's findings with the reports of bi/multinucleate melanocytes in zebrafish as seen in earlier normal development Saunders et al, *Thyroid hormone regulates distinct paths to maturation in pigment cell lineages*, 2019.

- Authors should better indicate what cell lines and variants are being used in the Figure title or legends (e.g. Fig 3, 4, 5 – all RPE-1-related lines).

- The paper uses both histograms and scatter plots, with scatter plots becoming more often required (may be needed per journal editorial policy?).

Reviewer #2 (Remarks to the Author):

In this manuscript, Darp et al. report that oncogenic BRAFV600E can induce whole-genome doubling in zebrafish melanocytes and human RPE1 cells through suppression of cytokinesis. Using a zebrafish model of p53-inactivated BRAF-mut melanoma, the authors found that Tg(mitfa:BRAFV600E); p53(lf) melanocytes were tetraploid and binucleate. Tetraploidy was induced in RPE1 human cells following the overexpression of mutant (but not WT) BRAF, through

failed cytokinesis. Looking specifically into known cytokinesis players, the authors found activation of Rac1 and reduced RhoA localization to the cell equator in BRAFV600E-expressing cells, which could be reversed by MEK or ERK inhibition. RAF, MEK and ERK inhibitors specifically reduced tetraploidy formation when cells were exposed to them during late G1 and early S phases. An increase in supernumerary centrosomes was observed in BRAFV600E-expressing cells, which could also be suppressed by MEK or ERK inhibition. The suppression of supernumerary centrosomes partially rescued RhoA localization and cytokinesis in BRAF-mut cells. Finally, the manuscript demonstrates that BRAFV600E-induced tetraploid cells arrest in G1, and that this arrest is alleviated by loss of p53, suggesting that p53 inactivation is necessary for the proliferation of BRAFV600E-induced tetraploid cells.

Overall, this is an interesting study, which includes logical and well-controlled experiments. The combination of both an *in vivo* zebrafish model and an *in vitro* human cell line model is a plus. Given that the link between BRAFV600E and supernumerary centrosomes in melanoma has been previously described, much of the novelty in the paper lies in its mechanistic studies. However, these studies are focused on pre-determined players, which may lead to a partial picture/model of the mechanism that the paper is trying to decipher. While I generally support the publication of this manuscript in *Nat Commun*, I believe that it could benefit from expanding the scope of its mechanistic studies. Below are specific comments.

1) The manuscript provides some evidence for the involvement of the MAPK pathway in the observed link between BRAF mutation, supernumerary chromosome and WGD: that mutant NRAS led to a similar increase in tetraploidy formation; and that MEK and ERK inhibitors restored RhoA localization and reduced tetraploidy formation. A few questions remain, however:

- a) Why would the "paradox-breaking" drugs achieve the same effect as the MEK and ERK inhibitors?
- b) Is there a role for the previously-reported effect of mutant BRAF on the spindle assembly checkpoint activity (and MPS1 in particular)?
- c) Also, does the interaction between the mutant BRAF and CRAF play a role in the induction of WGD, given that in some contexts BRAF was shown to drive aneuploidy by deregulating CRAF, and that CRAF was shown to regulate Rho signaling?
- d) Finally, is there a role for MEK-independent BRAF/CRAF activity, which was previously reported to affect mitosis and tumor progression? Potentially interesting players here may be AURKA and PLK1.

2) The authors followed G1 tetraploids after release from synchronization to assess cell cycle progression, showing that p53 inactivation was necessary for the proliferation of the tetraploid cells. Can they culture the cells for longer time periods after tetraploidy induction, and quantify the proportion of tetraploid cells in the population in p53-WT vs. p53-null cells? If p53 inactivation is necessary for the proliferation of the tetraploid cells, then dox withdrawal should lead to a fast decrease in the prevalence of tetraploid cells, in a p53-dependent manner. In other words, some kind of a competition assay would be beneficial here in order to drive this important point home.

3) Is there an association between WGD and BRAF mutations and/or RAS/MAPK activation in human tumors (e.g., in TCGA data)? Exploring the existing genomic data from tumors would help demonstrate the relevance of the findings to tumor biology, and may also help determine the MAPK-dependent vs. MAPK-independent effects (related to the first point).

Reviewer #3 (Remarks to the Author):

In this manuscript by Darp et al, the authors explore the relationship between oncogenic BRAF mutations frequently found in melanoma and another feature frequently found in this tumor, namely whole genome duplication. The authors start off by analyzing the DNA content and nuclear size of melanocytes in tumors induced in a zebrafish BRAFV600E melanoma model, determining that the vast majority of the tumor cells in this model is tetraploid and binucleated (fig 1). To better understand the relationship between oncogenic BRAF and this phenotype, the authors generate a sophisticated human RPE1 cell derivative with a Fucci reporter and a cassette inducibly

expressing oncogenic BRAF. Based on this model, the authors convincingly show that the RPE1 cells exposed to acute expression of oncogenic BRAF fail cytokinesis, providing a mechanistic explanation for the phenotype described in vivo (fig. 2). In figure 3, the authors show elegant synchronization combined with staining for Anilin and RhoA in the first mitosis following expression of oncogenic BRAF. Based on this assay, the authors convincingly show that the oncogenic BRAF greatly affects the localization of those key cytokinesis regulators (fig. 3). Somewhat surprisingly, this aberrant mitotic activity relies on hyperactivation of MEK/ERK taking place before cells entered S/G2/M phase (fig. 4). Given that the lack of cytokinetic RhoA activity/localization could be suppressed by inhibition of Rac1 and that Rac1 activity can be promoted by increased microtubule nucleating capacity caused by supernumerary centrosomes, the authors went on to measure the number of mitotic centrioles observed upon oncogenic BRAF expression. Based on such counting (fig. 5), on the fact that extra centrioles negatively correlated with RhoA mislocalization and on the fact that PLK4 (a key centriole biogenesis regulator) inhibition restored RhoA localization (fig. 6), the authors conclude that cytokinesis failure promoted by oncogenic BRAF depends, at least in part, on the presence of supernumerary centrosomes. Finally, the authors provide experimental evidence supporting the notion that genome duplication promoted in tumors driven by oncogenic BRAF is already present in the tumor cell of origin.

The topic covered by this paper is certainly of interest for a broad readership spanning from cell biologists to cancer researchers. The use of transgenic animal models combined with state-of-the-art imaging in human cell lines can be certainly seen as elements supporting publication in Nat Comms. However, the paper has in my opinion two main shortcomings that need to be satisfactorily addressed before publication:

1) the proposed series of events starting from oncogenic BRAF expression, appearance of supernumerary centrioles/centrosomes, increased Rac1 activity leading to RhoA deregulation and resulting into cytokinesis failure is not supported by the data. First of all, the penetrance of the presence of supernumerary centrioles in mitosis is much lower than the percentage of cytokinesis failure in the same mitosis. This already shows, as the authors point out, that the centrosomal problem can account at best in part for the cytokinetic phenotype. More importantly, while it is plausible that oncogenic BRAF expression in G1 can lead to the presence of extra centrioles in the subsequent mitosis, how can this lead to hyperactivation of Rac1 is extremely puzzling to me. In fact, newborn centrioles (also known as procentrioles) need to pass at least one mitotic division to become competent to recruit pericentriolar material and thereby to nucleate microtubules. If oncogenic BRAF was able to generate MTOC competent centrioles in the same interphase (i.e. without mitotic traverse) it would be something extremely noteworthy. At the present stage and based on the data presented, I think that this claim cannot be made. If the mitotic defects at the RhoA levels are not caused by supernumerary centrosomes, it remains completely obscure how can the BRAFV600E mutation promote this mitotic defect through its G1 activity.

2) in the last figure of the paper (7), the authors show that the presumed melanoma cell of origin is tetraploid in the zebrafish oncogenic BRAF model. The same cell is, however, mononucleated. This datum argues that either cytokinesis failure has not yet happened at this stage, or that following an abortive cytokinesis, the melanoma cell of origin has undergone an additional cell cycle culminating with a complete karyokinesis AND cytokinesis. How this can be reconciled with the extremely high penetrance of binucleated cells in the melanomas shown in fig.1 remains to be clarified.

Additional issues:

a) Page 6 "... melanocytes in these dorsal regions were larger in size and fewer in number". The data need to be quantified

b) Fig. 1E e 1G are redundant. Move 1G to supplementary?

c) Supplementary Fig. 1B: from this image it is hard to believe that those nuclei belong to the same cell. Please show a better image, comparable to Fig 1D or Suppl Fig 1A and with the same scale bar.

d) Page 11: "...p53(lf) melanocytes were much larger than those of Tg(mitfa:BRAFV600E) melanocytes (Fig. 7A)". Please quantify the size.

e) Page 12: "As compared to parental BRAFV600E-inducible RPE1-FUCCI cells, BRAFV600E-inducible P53^{-/-} RPE-1 FUCCI cells had a lower fraction of G1 tetraploids following BRAFV600E expression (Supplementary Fig. 5F, G), consistent with the notion that an arrest of G1 tetraploids was bypassed in P53^{-/-} cells". A decrease in G1 tetraploids in the p53^{-/-} cell line is interpreted as a bypass of p53-dependent cell cycle arrest. If this were the case, a concomitant increase in 8N cells should be seen. In general, looking at a proliferation marker in the different ploidy compartments would seem more appropriate to infer the presence/absence of a cell cycle arrest.

f) another paper has reported that oncogenic BRAF promotes the formation of extra centrosomes, PMID: 20068179. This paper should be cited and discussed appropriately.

POINT-BY-POINT RESPONSE TO REVIEWERS

We thank the reviewers for their comments. Below we address every concern expressed by the reviewers point-by-point. In most cases, we have included new data below and refer to any changes that have been made to the manuscript. Within the manuscript itself, changes are highlighted in yellow to facilitate re-review.

Addressing all of these concerns has substantially improved our manuscript. There is now data supporting conservation of the findings to a human melanocyte system to complement the findings in our zebrafish model. There is additional mechanistic insight into the connection between supernumerary centrioles and cytokinesis failure. We also clarified the effect of TP53 on the progression of cells that experience cytokinesis failure. We hope that you find the revised manuscript suitable for publication in Nature Communications.

Reviewer #1 (Remarks to the Author):

In “Oncogenic BRAF Induces Whole-Genome Doubling Through Suppression of Cytokinesis” by Darp et al, the authors use in vivo zebrafish melanoma models and tractable in vitro cell line models to determine the role of overactive BRAF in disrupting the normal processes of cell division/RhoA function and contributing to the abnormal growth/behavior of melanoma cancer. The study addresses interesting questions that touch on important fundamental aspects of cancer biology related to the precise mechanisms of how specific oncogenic events fan out to disrupt normal cell biology and has specific relevance to the relationship between preneoplastic melanocytes, nevi, and frank melanomas. The experiments are overall well done and generally well explained with appropriate conclusions drawn, which will likely be of interest to a wide audience. With a few important weaknesses addressed (below), the manuscript would be appropriate for publication:

Major points:

1) For this reviewer, the use of model organisms (zebrafish) and cell lines (RPE-1 and related) is reasonable and appropriate. However, the authors should better describe the RPE-1 line (presumably TERT immortalized, as in Shenk et al, Method Mol Biol, 2016?), and the relationship of this cell type to normal melanoma formation. RPE is a pigmented epithelium but not the precursor to cutaneous melanoma – which is okay for the elegant and precise cell cycle, etc. analysis which is made feasible with the RPE-1 FUCCI and related engineered lines – but this should be better highlighted and more openly explained in results/discussion. Related - might TERT-mediated immortalization of RPE-1 cells have any effect on the underlying cell biological/cytokinesis mechanisms at play?

• We appreciate the reviewer’s comments on the appropriateness of our combined use of a model organism and cell line. To better explain the RPE-1 cell line we used, we have added text in our Results section stating “RPE-1 cells were chosen for this analysis because they are non-transformed, hTERT-immortalized, and have a stable, diploid karyotype.”

• Related to the point that RPE-1 cells are not precursors of melanoma, we have broadened our analyses to the Mel-ST immortalized melanocyte cell line. As described below, BRAFV600E-dependent RhoA activation and centriole amplification were also observed in Mel-ST cells, indicating that the results are pertinent to melanocyte biology.

• Related to a potential effect of hTERT-mediated immortalization on the underlying cell biology, there are a couple of notable considerations. Firstly, similar cell biological effects of BRAFV600E expression are observed in zebrafish cells that do not express telomerase, which indicates the cell biological phenotypes are not hTERT-dependent. Secondly, we consider hTERT-immortalized cells to be superior for in vitro analyses because such cells are less prone to mitotic defects than cells immortalized through other means. hTERT-immortalized cells avoid telomere crisis and resulting chromosome bridges, which are associated with cytokinesis failure (Reviewed in Mierzwa et al., Dev Cell, 2014, PMID: 25490264 and Lens et al., Nat Rev Cancer, 2019, PMID: 30523339).

2) The authors note that supernumerary centrosomes have been noted in established melanoma cell lines and in patient samples, supporting the “in human disease” relevance of the findings. However, without repeating the elegant cell cycle analyses made possible with the RPE-1 FUCCI line and related engineered lines in a melanocyte, the paper would seem to still implicitly argue that BRAF-activation would lead to RhoA mislocalization and accumulation of centrioles soon/immediately after induction of BRAFV600E in a melanocyte system. Such an

experiment, which should be feasible with RhoA and Centrin staining in BRAFV600E expressing melanocytes, would strengthen/bridge the RPE-1 cell line work and “melanoma tumor initiation from a melanocyte” context which the paper seems targeted towards.

It is important to show relevance of our findings in a melanocyte system. To do this we measured RhoA activity and centriole numbers in immortalized melanocyte Mel-ST cells. For RhoA we quantified activity using western blots for RhoA-GTP as compared to total RhoA. With this assay we found that RhoA activity was substantially reduced in BRAFV600E-expressing Mel-ST cells as compared to control Mel-ST cells. These data are included in our companion manuscript (Vittoria et al.) that is concurrently in revision for Nature Communications. We also quantified centrioles by Centrin staining in Mel-ST cells and found an increase in supernumerary centrioles upon BRAFV600E expression. The Centrin staining data have been added into a new Supplementary Figure 5.

RhoA-GTP quantification from Mel-ST cells:

[REDACTED]

Centriole quantification using Centrin staining from Mel-ST cells:

Minor points:

- Should cite Santoriello et al., “Kita Driven Expression of Oncogenic HRAS Leads to Early Onset and Highly Penetrant Melanoma in Zebrafish” in PLOS One (2010) as having seen

binucleate melanocyte in this different melanoma model – seems to support the even more broad applicability of these proposed mechanisms to another oncogenic driver. Also, while almost certainly a different phenomenon, worth further contextualizing the paper’s findings with the reports of bi/multinucleate melanocytes in zebrafish as seen in earlier normal development Saunders et al, Thyroid hormone regulates distinct paths to maturation in pigment cell lineages, 2019.

Thank you for the suggested references. We have now included them in our revised manuscript and agree that they help to add context to our findings. For the Santoriello study we have included the following text in our Results section: “Melanocytes from animals expressing an NRAS^{Q61L} oncogene that is commonly found in human melanomas were also binucleate (Fig. 1E, Supplementary Fig. 1C), and binucleation has also been observed in zebrafish melanocytes that express an oncogenic variant of HRAS.” For the Saunders and one other study of binucleation in normal melanocytes (Usui et al., Dev Growth Differ, 2018, PMID: 30088265) we have added the following text in our Results section: “Whereas binucleate dorsal epidermal melanocytes are rare in wild-type animals, binucleate stripe-associated dermal melanocytes are more commonly observed, suggesting that mechanisms that coordinate nuclear and cellular divisions may be particularly prone to regulation in zebrafish melanocytes.”

- Authors should better indicate what cell lines and variants are being used in the Figure title or legends (e.g. Fig 3, 4, 5 – all RPE-1-related lines).

We have now specified the cell lines and variants used in all figure legends.

- The paper uses both histograms and scatter plots, with scatter plots becoming more often required (may be needed per journal editorial policy?).

We appreciate the reviewer for pointing this out. We have modified bar graphs so they are overlaid with scatter plot data, and they now conform to Nature Communications guidelines for data presentation. We also have changed the graph of melanocyte density in Figure 1C from a bar graph to a scatter plot.

Reviewer #2 (Remarks to the Author):

In this manuscript, Darp et al. report that oncogenic BRAFV600E can induce whole-genome doubling in zebrafish melanocytes and human RPE1 cells through suppression of cytokinesis. Using a zebrafish model of p53-inactivated BRAF-mut melanoma, the authors found that Tg(mitfa:BRAFV600E); p53(lf) melanocytes were tetraploid and binucleate. Tetraploidy was induced in RPE1 human cells following the overexpression of mutant (but not WT) BRAF, through failed cytokinesis. Looking specifically into known cytokinesis players, the authors found activation of Rac1 and reduced RhoA localization to the cell equator in BRAFV600E-expressing cells, which could be reversed by MEK or ERK inhibition. RAF, MEK and ERK inhibitors specifically reduced tetraploidy formation when cells were exposed to them during late G1 and early S phases. An increase in supernumerary centrosomes was observed in BRAFV600E-expressing cells, which could also be suppressed by MEK or ERK inhibition. The suppression of supernumerary centrosomes partially rescued RhoA localization and cytokinesis in BRAF-mut cells. Finally, the manuscript demonstrates that BRAFV600E-induced tetraploid cells arrest in G1, and that this arrest is alleviated by loss of p53, suggesting that p53 inactivation is necessary for the proliferation of BRAFV600E-induced tetraploid cells.

Overall, this is an interesting study, which includes logical and well-controlled experiments. The combination of both an in vivo zebrafish model and an in vitro human cell line model is a plus. Given that the link between BRAFV600E and supernumerary centrosomes in melanoma has been previously described, much of the novelty in the paper lies in its mechanistic studies. However, these studies are focused on pre-determined players, which may lead to a partial picture/model of the mechanism that the paper is trying to decipher. While I generally support the publication of this manuscript in Nat Commun, I believe that it could benefit from expanding the scope of its mechanistic studies. Below are specific comments.

1) The manuscript provides some evidence for the involvement of the MAPK pathway in the observed link between BRAF mutation, supernumerary chromosome and WGD: that mutant NRAS led to a similar increase in tetraploidy formation; and that MEK and ERK inhibitors restored RhoA localization and reduced tetraploidy formation. A few questions remain, however: a) Why would the “paradox-breaking” drugs achieve the same effect as the MEK and ERK inhibitors?

The paradox-breaking drugs are expected to have the same effect as MEK and ERK inhibitors since they also inhibit overall MAPK(ERK) pathway signaling activity (Supp. Fig. 3A, 3B). Perhaps this question is motivated by the ‘paradoxical RAF activation’ with vemurafenib, where at certain concentrations of vemurafenib, increased downstream MAPK(ERK) pathway activity is observed in treated cells. This effect is cell-type dependent, concentration-dependent and also exacerbated in cells that contain activating RAS mutations (Holderfield et al., BJC, 2014 PMID: 24642617; Poulidakos et al., Nature, 2010, PMID: 20179705). In our case, the concentration of vemurafenib we are using in RPE-1 cells profoundly inhibits downstream MAPK(ERK) activity, so there is no paradoxical activation under the conditions we used, and we do not expect results from vemurafenib treatment to differ from treatment with MEK inhibitor, ERK inhibitor, paradox-breaking BRAF inhibitors or any drug that inhibits MAPK(ERK) pathway activity.

b) Is there a role for the previously-reported effect of mutant BRAF on the spindle assembly checkpoint activity (and MPS1 in particular)?

To address this question we measured levels of total MPS1 and MPS1 phosphorylated on T676, the T loop residue whose phosphorylation is critical for activation. Levels of MPS1 and phospho-MPS1 (by western blot) were unchanged upon BRAF expression in RPE-1 FUCCI cells. We also stained cells for spindle checkpoint proteins MAD1 (by IF) and MAD2 (by western blot) and found that levels of each protein were unchanged in BRAFV600E-expressing cells.

We are aware that BRAFV600E expression in melanoma cells has been shown to increase MPS1, MAD1 and MAD2 levels (Cui et al., *Oncogene*, 2008, PMID: 18071315). There are several differences between the system we are using and the systems used in these previous experiments that could account for these incongruent results. These differences include, but are not limited to: a) Cell type – we performed our experiments in non-transformed RPE-1 cells and previous experiments were performed in melanoma cell lines. Thus, melanoma cell-specific dysregulation of the cell cycle could be necessary for, or help to uncover, BRAFV600E effects on the spindle checkpoint, b) Cell ploidy – RPE-1 cells are diploid, whereas SK-MEL5 cells showing BRAFV600E regulation of MPS1, MAD1, MAD2 are hyperpentaploid. The aneuploidy in these cells may provide a sensitized background in which spindle checkpoint regulation by BRAFV600E could be uncovered but would not be evident in diploid RPE-1 cells, c) The synchronization procedure we used focused analysis on proximate BRAFV600E effects when BRAFV600E is expressed in a specific cell cycle window, i.e. effects that occur in the cell cycle immediately following BRAFV600E induction. Previous experiments linking BRAFV600E with MPS1 and checkpoint protein changes were performed in unsynchronized cells when BRAFV600E was expressed throughout the cell cycle. This broader expression could reflect links that would occur outside of the scope of our analyses.

c) Also, does the interaction between the mutant BRAF and CRAF play a role in the induction of WGD, given that in some contexts BRAF was shown to drive aneuploidy by deregulating CRAF, and that CRAF was shown to regulate Rho signaling?

The regulation of Rho by CRAF has been shown to occur through the Rho effector Rok-alpha (Ehrenreiter, et al., *JCB* 2005, PMID: 15753127 and Varga, et al., *SciSig*, 2017, PMID: 28270557), and this regulation acts in parallel to MEK signaling. Since BRAFV600E-dependent downregulation of RhoA was entirely reversed by a MEK or ERK inhibitor (Figure 3D), we surmise that the effect of BRAFV600E on RhoA is predominantly through MEK/ERK signaling. As described immediately below, we did investigate other MEK-independent effects of RAF signaling, but we did not see any evidence of other MEK-independent effectors that are important for the BRAFV600E-dependent effects in our studies.

d) Finally, is there a role for MEK-independent BRAF/CRAF activity, which was previously

reported to affect mitosis and tumor progression? Potentially interesting players here may be AURKA and PLK1.

The involvement of MEK-independent BRAF/CRAF activity is potentially interesting. Overexpression of PLK1 or AURKA causes failure of cytokinesis (Mundt et al., BBRC, 1997, PMID: 9344838; Meraldi et al., EMBO J, 2002, PMID: 11847097), and it has been shown that CRAF can associate with these proteins at centrosomes in G2/M cells (Mielgo et al., Nat Med, 2011, PMID: 22081024). Thus, it is conceivable that BRAFV600E-dependent activation of CRAF could lead to activation of PLK1 or AURKA, ultimately causing failure of cytokinesis. To test this possibility we applied commercially-available inhibitors of PLK1 and AURKA to BRAFV600E-expressing cells and determined effects on G1 tetraploids. As shown and discussed below, we found no evidence of PLK1 or AURKA involvement.

In the above experiments, RPE1-FUCCI cells were synchronized and treated with Dox to induce BRAFV600E as normal. The PLK1 inhibitor volasertib and AURKA inhibitor alisertib were applied to cells coincident with Dox and kept on throughout the course of the experiment. The PLK1 inhibitor had no significant effect on (i.e. did not suppress) the BRAFV600E-driven formation of G1 tetraploids. The AURKA inhibitor had profound effects on cytokinesis, leading to an 8-fold increase in G1 tetraploids upon treatment with inhibitor alone. Concurrent BRAFV600E expression and AURKAi treatment was not significantly different than treatment with AURKAi alone. This could be interpreted in at least two different ways: 1) BRAFV600E expression and AURKAi treatment is not additive, possibly indicating suppression of BRAFV600E increase by AURKAi, or 2) AURKAi treatment causes the maximum amount of G1 tetraploid formation and concurrent BRAFV600E expression and AURKAi treatment could not go higher, thus making it difficult to interpret the results. Neither interpretation is satisfying, and more to the point, the increase in G1 tetraploids caused by AURKAi treatment defeats the design of the experiment and makes it not suitable to include in our revision.

NB: In Figure 4A in our manuscript we treated cells with MEK and ERK inhibitors and found that nearly all of the excess G1 tetraploid formation caused by BRAFV600E was dependent on MEK/ERK activity. There may be a minor contribution of MEK/ERK-independent activity. Our results above indicate any such minor contribution does not depend on PLK1.

2) The authors followed G1 tetraploids after release from synchronization to assess cell cycle progression, showing that p53 inactivation was necessary for the proliferation of the tetraploid

cells. Can they culture the cells for longer time periods after tetraploidy induction, and quantify the proportion of tetraploid cells in the population in p53-WT vs. p53-null cells? If p53 inactivation is necessary for the proliferation of the tetraploid cells, then dox withdrawal should lead to a fast decrease in the prevalence of tetraploid cells, in a p53-dependent manner. In other words, some kind of a competition assay would be beneficial here in order to drive this important point home.

In considering this experiment, it is important to specify predictions for different populations of cells. For G1 tetraploids, those that are generated following BRAFV600E expression, we expect G1 tetraploids in a p53-WT background to be arrested even after Dox is withdrawn. For G1 tetraploids in a p53-null background, we expect G1 tetraploids to be lower after Dox withdrawal, as they are in the experiment we conducted in the original Supplementary Figure 5F (now Supplementary Figure 6G). For S/G2/M cells, the relevant populations are those that are >4N, which would be derived from G1 tetraploid cells that cycled into S/G2/M phases. We expect to find virtually none of these cells in p53-WT cells since their precursor G1 tetraploids are arrested. In p53-null cells we expect to see some percentage of >4N S/G2/M cells immediately following Dox withdrawal. After a long period of culture we expect such cells to be present because, assuming they continued to cycle, they would not experience a G1 arrest in the p53-null background.

As requested by the reviewer we conducted this experiment, in which we synchronized cells and induced BRAFV600E expression as usual, then withdrew Dox and cultured cells for 4 days. Using flow cytometry we measured G1 tetraploids and >4N S/G2/M cells. One day following Dox withdrawal, our results were similar to those we observed in the original Supplementary Figure 5F (now Supplementary Figure 6G): in the p53-WT background there were more G1 tetraploid cells, and in the p53-null background there were fewer G1 tetraploid cells. We also found that in the p53-WT background there were fewer >4N S/G2/M cells, whereas such cells were more abundant in the p53-null background. After 4 days in the p53-WT background there were no >4N S/G2/M cells. After 4 days in the p53-null background >4N S/G2/M cells were present, although not as many as in the p53-null culture in the day following BRAFV600E induction. Taken together, these results suggest that BRAFV600E expression leads to formation

of tetraploid cells, and such cells arrest in a p53-WT background and remain arrested if cultures are grown longer-term. The results also suggest that BRAFV600E expression causes formation of G1 tetraploids, and some of these cells are able to progress into S/G2/M. In longer-term p53-null cultures >4N S/G2/M cells are present, suggesting that tetraploid cells, once generated, can continue to cycle through multiple divisions. These new data are shown below and have been included as panels H, I and J in an expanded Supplementary Figure 6.

3) Is there an association between WGD and BRAF mutations and/or RAS/MAPK activation in human tumors (e.g., in TCGA data)? Exploring the existing genomic data from tumors would help demonstrate the relevance of the findings to tumor biology, and may also help determine the MAPK-dependent vs. MAPK-independent effects (related to the first point).

We have analyzed the association of WGD with BRAF mutations as well as ANY RAS/MAPK pathway genes (BRAF, NRAS, KRAS, HRAS, NF1, MAP2K1, MAP3K13 and PTPN11). We have also analyzed the association of WGD with BRAF or ANY RAS/MAPK mutations together with TP53 or CDKN2A mutations with the rationale that our data support a combination of RAS/MAPK activation plus TP53 pathway inactivation enable progression of cells that undergo WGD. The genomic data used were from two large, independent cohorts: a) the TCGA cohort (<https://www.cancer.gov/tcga>), and b) the MSK-IMPACT cohort (Zehir et al., Nature Medicine, 2017, PMID:28481359), both of which include many tumor types. Each cohort was previously analyzed for WGD (TCGA: Taylor et al., Cancer Cell, 2018, PMID:29622463; MSK-IMPACT: Bielski et al., Nature Genetics, 2018, PMID:30013179). Below we have summarized our analyses and have included graphs to visualize the findings and an attached spreadsheet with detailed analyses.

We conducted analyses that could reveal any trends that were consistent across both cohorts. There are several notable observations from these analyses (for points A, D, and E, there relevant comparisons are indicated in the graphs immediately below):

- A) Although there appears to be a negative correlation between BRAF and ANY RAS/MAPK mutations with WGD in the TCGA cohort, this negative correlation is not observed in the MSK-IMPACT cohort (see below and attached). The reason for this difference is not clear but could be related to different sensitivity and mutation calling used for the two different cohorts.*
- B) As has been previously discovered (Bielski et al., Nature Genetics, 2018, PMID:30013179), there is a strong positive correlation between mutations in TP53 and WGD, and we saw this correlation in both cohorts (see attached).*
- C) We analyzed CDKN2A mutations because they also disable TP53 signaling and are commonly observed in melanomas. In both cohorts we found a weak positive correlation between CDKN2A loss and WGD, suggesting that TP53 pathway loss is important for WGD (see attached). For this reason we considered tumors that had either TP53 or CDKN2A loss as one group.*
- D) The prevalence of WGD in TP53 or CDKN2A mutant tumors depended very little on the presence of a BRAF (TCGA: $P=0.27$; MSKI: $P=0.98$) or ANY RAS/MAPK mutation (TCGA: $P=0.04$; MSKI: $P=0.88$) (see below and attached). The presence of ANY RAS/MAPK mutation in TP53 or CDKN2A mutant tumors in the TCGA cohort was statistically significant, but this was not observed in the MSK-IMPACT cohort.*
- E) In tumors that did not have a TP53 or CDKN2A mutation, there was a substantial difference between tumors that had a BRAF mutation and those that did not (see below and attached), with the presence of a BRAF mutation being associated with less WGD ($P<0.0001$). The same trend was observed with ANY RAS/MAPK ($P<0.0001$). Although speculative, it is possible that this lower percentage of WGD results from detrimental*

effects of a BRAF or ANY RAS/MAPK mutation in the absence of a TP53 or CDKN2A mutation. However, in each of these cases the same trend was not statistically significant in the MSK-IMPACT cohort ($P=0.10$ and $P=0.09$).

Taken together, these data can be interpreted as follows: 1) TP53 pathway loss is a major determinant for WGD, with TP53 mutations showing the strongest correlation with WGD, 2) since the correlation between TP53/CDKN2A and WGD is independent of BRAF or ANY RAS/MAPK mutation it could reflect that TP53 pathway loss is a rate limiting step in progression of WGD cells, 3) many TP53/CDKN2A mutant tumors exhibit WGD without being mutant for BRAF or ANY RAS/MAPK gene, suggesting that there are means other than BRAF or ANY RAS/MAPK mutation by which tumors can undergo WGD, 4) as seen in the TCGA cohort, the negative correlation of BRAF and ANY RAS/MAPK mutations with WGD in the absence of TP53/CDKN2A mutations could reflect a block in cell cycle progression that occurs when BRAF or ANY RAS/MAPK activation occurs with an intact TP53 pathway. Overall, the data suggest that a substantial fraction of BRAF or ANY RAS/MAPK mutant tumors exhibit WGD, and WGD in these tumors may be enabled by TP53 pathway loss, but BRAF or ANY RAS/MAPK mutations do not appear to alone drive WGD. Because the last three of the points above (2, 3 and 4) are speculative and the data – except for the role of TP53 – are not statistically significant across both cohorts, we do not feel this analysis informs the current study and have chosen to not include it in the manuscript.

Reviewer #3 (Remarks to the Author):

In this manuscript by Darp et al, the authors explore the relationship between oncogenic BRAF mutations frequently found in melanoma and another feature frequently found in this tumor, namely whole genome duplication. The authors start off by analyzing the DNA content and nuclear size of melanocytes in tumors induced in a zebrafish BRAFV600E melanoma model, determining that the vast majority of the tumor cells in this model is tetraploid and binucleated (fig 1). To better understand the relationship between oncogenic BRAF and this phenotype, the authors generate a sophisticated human RPE1 cell derivative with a Fucci reporter and a cassette inducibly expressing oncogenic BRAF. Based on this model, the authors convincingly show that the RPE1 cells exposed to acute expression of oncogenic BRAF fail cytokinesis, providing a mechanistic explanation for the phenotype described in vivo (fig. 2). In figure 3, the authors show elegant synchronization combined with staining for Anilin and RhoA in the first mitosis following expression of oncogenic BRAF. Based on this assay, the authors convincingly show that the oncogenic BRAF greatly affects the localization of those key cytokinesis regulators (fig. 3). Somewhat surprisingly, this aberrant mitotic activity relies on hyperactivation of MEK/ERK taking place before cells entered S/G2/M phase (fig. 4). Given that the lack of cytokinetic RhoA activity/localization could be suppressed by inhibition of Rac1 and that Rac1 activity can be promoted by increased microtubule nucleating capacity caused by supernumerary centrosomes, the authors went on to measure the number of mitotic centrioles observed upon oncogenic BRAF expression. Based on such counting (fig. 5), on the fact that extra centrioles negatively correlated with RhoA mislocalization and on the fact that PLK4 (a key centriole biogenesis regulator) inhibition restored RhoA localization (fig. 6), the authors conclude that cytokinesis failure promoted by oncogenic BRAF depends, at least in part, on the presence of supernumerary centrosomes. Finally, the authors provide experimental evidence supporting the notion that genome duplication promoted in tumors driven by oncogenic BRAF is already present in the tumor cell of origin.

The topic covered by this paper is certainly of interest for a broad readership spanning from cell biologists to cancer researchers. The use of transgenic animal models combined with state-of-the-art imaging in human cell lines can be certainly seen as elements supporting publication in Nat Comms. However, the paper has in my opinion two main shortcomings that need to be satisfactorily addressed before publication:

1) the proposed series of events starting from oncogenic BRAF expression, appearance of supernumerary centrioles/centrosomes, increased Rac1 activity leading to RhoA deregulation and resulting into cytokinesis failure is not supported by the data. First of all, the penetrance of the presence of supernumerary centrioles in mitosis is much lower than the percentage of cytokinesis failure in the same mitosis. This already shows, as the authors point out, that the centrosomal problem can account at best in part for the cytokinetic phenotype. More importantly, while it is plausible that oncogenic BRAF expression in G1 can lead to the presence of extra centrioles in the subsequent mitosis, how can this lead to hyperactivation of Rac1 is extremely puzzling to me. In fact, newborn centrioles (also known as procentrioles) need to pass at least one mitotic division to become competent to recruit pericentriolar material and thereby to nucleate microtubules. If oncogenic BRAF was able to generate MTOC competent centrioles in the same interphase (i.e. without mitotic traverse) it would be something extremely noteworthy. At the present stage and based on the data presented, I think that this claim cannot be made. If the mitotic defects at the RhoA levels are not caused by supernumerary centrosomes, it remains completely obscure how can the BRAFV600E mutation promote this mitotic defect through its G1 activity.

These are excellent points. One thing to clarify before we address these concerns – the penetrance of extra centrioles is higher as compared to cytokinesis failure in the same mitosis; we found that ~20% of cells have extra centrioles in the mitosis following BRAFV600E expression (Fig. 5B), whereas the percent of cells experiencing cytokinesis failure in this same mitosis was lower (~11% if measured as G1 tetraploids in Fig. 2C, or ~8% if measured as cytokinesis failure in live cell imaging in Fig. 2F). Nonetheless, as the reviewer states, our data overall support the conclusion that centrosomal defects underlie only some portion of cytokinesis defects, which is what we intended to convey in our manuscript. To address the reviewer’s main point, i.e., the maturation state of supernumerary centrioles that would affect their ability to activate Rac1, we asked whether cells with BRAFV600E-induced supernumerary centrioles showed mitotic spindles in which microtubule nucleation to at least two centrioles or centriole pairs at the same spindle pole was evident. In immunofluorescence with anti-Centrin2 and anti-Alpha Tubulin antibodies, we found microtubule nucleation to at least two centrioles or centriole pairs at the same pole was observed in a majority of cases. These data are shown below and have been included in a revised Supplementary Figure 4. In addition, we performed immunofluorescence with anti-Pericentrin antibodies to assess pericentriolar material (PCM) expansion on supernumerary centrioles. In mitoses with at least three separated centrioles or centriole pairs, we found that most showed Pericentrin positivity at all centrioles or centriole pairs (see data below which has been included in Supplementary Figure 4). These data suggest that supernumerary centrioles resulting from BRAFV600E expression had undergone maturation and were capable of nucleating microtubules.

As the reviewer indicates, this is notable. How precocious maturation of BRAFV600E-induced supernumerary centrioles occurs is not entirely clear, although a combination of studies suggest an interesting hypothesis. RAS/MAPK pathway activation, such as that induced by BRAFV600E, is a well-established cause of replication stress (DiMicco et al., Nature, 2006, PMID:17136094; reviewed by Grabocka et al., Clinical Cancer Research, 2015,

PMID:25424849). Recently, it has been shown that replication stress causes premature centriole disengagement, leading to the formation of multipolar spindles that show microtubule nucleation on prematurely disengaged centrioles (Wilhelm et al., Nature Communications, 2019, PMID:31395887). Premature centriole disengagement caused by disruption of Cep57 and PCM components also leads to precocious maturation, suggesting that premature disengagement can generally enable precocious maturation and MTOC activity (Watanabe et al., Nature Communications, 2019, PMID:30804344). Thus, taken together, RAS/MAPK-induced replication stress caused by BRAFV600E expression could lead to premature centriole disengagement followed by precocious maturation and MTOC activity.

We have included these points in our Discussion section. We have initiated follow up studies to further probe this hypothesis, which we respectfully believe are distinctly focused and extensive enough to go beyond the scope of the current manuscript.

2) in the last figure of the paper (7), the authors show that the presumed melanoma cell of origin is tetraploid in the zebrafish oncogenic BRAF model. The same cell is, however, mononucleated. This datum argues that either cytokinesis failure has not yet happened at this stage, or that following an abortive cytokinesis, the melanoma cell of origin has undergone an additional cell cycle culminating with a complete karyokinesis AND cytokinesis. How this can be reconciled with the extremely high penetrance of binucleated cells in the melanomas shown in fig.1 remains to be clarified.

We strived to be as clear as possible in our initial manuscript, but this is a point is where we could use some improvement. In our revised manuscript we have now clarified that the cells studied in Figure 7E are nascent tumor cells that are derived from a cell of origin. They themselves are not cells of origin; the lesions we studied in this panel were fewer than 20 cells but more than 10 cells. We have added in the lower end of 10 cells to make clear that these are not cells of origin. Another thing we have clarified is that the data in Figure 1, with the exception of panel A, pertain to melanocytes and not melanoma cells. The penetrance of binucleated cells is high in melanocytes that express BRAFV600E. However, binucleated melanoma cells are much less common in our zebrafish model. For these reasons we speculate that most cells that undergo cytokinesis failure arrest and do not progress to become melanomas. Loss of p53 function enables binucleated cells to enter the cell cycle and undergo DNA replication, but an additional block likely exists that prevents further progression through the cell cycle. As described in the third paragraph in our Discussion, we speculate that this additional block is bypassed in tumor cells of origin.

Additional issues:

a) Page 6 "... melanocytes in these dorsal regions were larger in size and fewer in number". The data need to be quantified.

We have quantified the size of melanocytes to show that melanocytes from Tg(mitfa:BRAFV600E) and Tg(mitfa:BRAFV600E); p53(lf) animals are larger than that of melanocytes from wild-type and p53(lf) animals. These data are shown below and are now included in a new panel in Supplementary Figure 1. For quantification of melanocytes, we could include a melanocyte count per scale but prefer a density measurement as it normalizes for differences in scale size. To more accurately reflect this choice, we have changed text in the relevant paragraph from melanocyte 'number' to 'density' and 'distribution'.

b) Fig. 1E e 1G are redundant. Move 1G to supplementary?

We appreciate the suggestion. We have moved Figure 1G to Supplementary Figure 1 and added the following sentence to the main text: "Flow cytometry also confirmed the binucleate nature of these cells (Supplementary Fig. 1D)."

c) Supplementary Fig. 1B: from this image it is hard to believe that those nuclei belong to the same cell. Please show a better image, comparable to Fig 1D or Suppl Fig 1A and with the same scale bar.

We have provided better images of the cell in Supplementary Figure 1B. The new images (immediately below) show the boundaries of the melanocyte based on melanin, the two Mitfa-positive nuclei that are bounded by melanin from this single cell, and other Mitfa-negative nuclei that serve as internal negative controls for Mitfa staining.

The images in Supplementary Figure 1B show the best example of a binucleate Tg(mitfa:NRASQ61K) melanocyte that we can obtain. Tg(mitfa:NRASQ61K) melanocytes are different from Tg(mitfa:BRAFV600E) and wild-type melanocytes in that they have a much higher density of melanin-containing melanosomes. This high density of melanosomes obscures nuclei in these cells, making it difficult to capture images of nuclei that are centrally-located in the cell body. Below is an example of a melanocyte with nuclei that are obscured by the high melanosome/melanin content. Thus, the melanocytes from these animals we could quantify and image were those with nuclei located near the periphery. We are confident that Tg(mitfa:NRASQ61K) cells, including the example shown above, were primarily binucleate because: a) the nuclei stain for Mitfa so are not from other cell types, b) the nuclei are present in a melanocyte that is not bordering or overlapping other melanocytes, and c) the two nuclei have

nearly identical morphologies that is common of the two nuclei in other binucleated melanocytes (e.g. F1D, F7A).

d) Page 11: "...nuclei of *Tg(mitfa:BRAFV600E); p53(lf)* melanocytes were much larger than those of *Tg(mitfa:BRAFV600E)* melanocytes (Fig. 7A)". Please quantify the size.

We have measured nuclear sizes (shown below) and included these data in Supplementary Figure 6.

e) Page 12: "As compared to parental BRAFV600E-inducible RPE1-FUCCI cells, BRAFV600E-inducible P53^{-/-} RPE-1 FUCCI cells had a lower fraction of G1 tetraploids following BRAFV600E expression (Supplementary Fig. 5F, G), consistent with the notion that an arrest of G1 tetraploids was bypassed in P53^{-/-} cells". A decrease in G1 tetraploids in the p53^{-/-} cell line is interpreted as a bypass of p53-dependent cell cycle arrest. If this were the case, a concomitant increase in 8N cells should be seen. In general, looking at a proliferation marker in the different ploidy compartments would seem more appropriate to infer the presence/absence of a cell cycle arrest.

We have addressed this question in a couple of ways. Firstly, we reanalyzed data from the experiment in which we isolated G1 tetraploids from BRAFV600E-expressing P53^{+/+} and P53^{-/-} RPE-1 FUCCI cultures (shown in Figure 7C) and replated them for further growth. After G1 tetraploids were replated we previously showed that only P53^{-/-} G1 tetraploids were able to progress into S/G2/M. We have analyzed the data to show that a large fraction of these cells are >4N, as would be predicted if G1 tetraploids progressed into S/G2/M. The new data are in the rightmost plot shown below and have been added into Figure 7C.

Secondly, we have performed an experiment to examine the presence of >4N S/G2/M cells in RPE-1 FUCCI cultures upon Dox-induced BRAFV600E expression and after Dox withdrawal. One day after withdrawal, G1 tetraploids were reduced in the P53^{-/-} culture and >4N S/G2/M cells were present. In the P53^{+/+} culture G1 tetraploids were present at higher levels and >4N S/G2/M cells were essentially absent. Longer-term culture of these P53^{-/-} cells showed that >4N S/G2/M cells remained present after 4 days. These data are shown below.

Taken together, these results show that, in P53^{-/-} cultures, a reduction in G1 tetraploids is accompanied by the presence and persistence of cells in S/G2/M with a >4N DNA content. The increased G1 tetraploids and absence of >4N S/G2/M cells indicates the failure of G1 tetraploids to progress into S/G2/M in these cultures.

f) another paper has reported that oncogenic BRAF promotes the formation of extra centrosomes, PMID: 20068179. This paper should be cited and discussed appropriately.

We appreciate the reviewer pointing out this reference. We had cited this paper in our Discussion along with another paper from the same group. The relevant text is: "Together these observations suggest that the defect caused by BRAFV600E is that of centrosomal overduplication. A similar phenotype has been observed upon BRAFV600E overexpression in established melanoma cell lines, although such cells had a high background of underlying centrosomal abnormalities^{68,69}."

MUTATION(S)	TCGA COHORT (9975 total tumors w/ sequencing and WGD data)						MSK-IMPACT COHORT (8190 total tumors w/ sequencing and WGD data)					
	NO. TUMORS	PERCENT w/ WGD	p (FISHER'S EXACT)	ODDS RATIO	OR (LOWER 95% CI)	OR (UPPER 95% CI)	NO. TUMORS	PERCENT w/ WGD	p (FISHER'S EXACT)	ODDS RATIO	OR (LOWER 95% CI)	OR (UPPER 95% CI)
ENTIRE COHORT	9975	35.8					8190	27.5				
BRAF	724	22.7	< 0.0001*	0.507	0.424	0.607	374	25.4	0.37	0.891	0.702	1.131
not BRAF	9251	36.8	< 0.0001*	1.971	1.649	2.357	7816	27.6	0.37	1.122	0.885	1.424
ANY RAS/MAPK	2436	31.6	< 0.0001*	0.781	0.709	0.861	2129	28.0	0.59	1.031	0.923	1.151
not ANY RAS/MAPK	7539	37.1	< 0.0001*	1.280	1.161	1.411	6061	27.4	0.59	0.970	0.869	1.083
TP53	3720	54.2	< 0.0001	3.599	3.301	3.924	3247	36.8	< 0.0001	2.131	1.931	2.351
BRAF + TP53	133	50.4	0.0005	1.839	1.306	2.589	131	34.4	0.093	1.384	0.962	1.991
ANY RAS/MAPK + TP53	869	49.6	< 0.0001	1.872	1.628	2.153	1006	35.6	< 0.0001	1.539	1.339	1.769
CDKN2A	1473	41.6	< 0.0001	1.338	1.195	1.498	1014	30.2	0.047	1.158	1.003	1.337
BRAF + CDKN2A	111	51.3	0.0009	1.91	1.313	2.778	77	37.7	0.054	1.597	1.004	2.538
ANY RAS/MAPK + CDKN2A	457	45.1	< 0.0001	1.503	1.244	1.815	369	34.3	0.0034	1.403	1.131	1.778
TP53 or CDKN2A	4496	49.9	< 0.0001	3.115	2.861	3.392	3796	34.9	< 0.0001	1.995	1.808	2.201
BRAF + (TP53 or CDKN2A)	215	47.9	0.0002	1.671	1.274	2.190	183	35.0	0.0292	1.427	1.049	1.941
TP53 or CDKN2A not BRAF	4281	50.0	< 0.0001	2.980	2.738	3.244	3613	34.9	< 0.0001	1.930	1.750	2.129
ANY RAS/MAPK + (TP53 or CDKN2A)	1131	46.1	< 0.0001	1.625	1.434	1.841	1179	35.1	< 0.0001	1.519	1.332	1.756
TP53 or CDKN2A not ANY RAS/MAPK	3365	51.1	< 0.0001	2.700	2.476	2.943	2617	34.8	< 0.0001	1.679	1.517	1.857
not TP53 or CDKN2A	5479	24.2	< 0.0001	0.321	0.295	0.350	4394	21.2	< 0.0001	0.501	0.454	0.553
BRAF not (TP53 or CDKN2A)	509	12.2	< 0.0001	0.236	0.180	0.309	191	16.2	0.0003	0.503	0.341	0.741
not TP53 or CDKN2A not BRAF	4970	25.4	< 0.0001	0.400	0.367	0.435	4203	21.4	0.0909	0.920	0.836	1.012
ANY RAS/MAPK not (TP53 or CDKN2A)	1305	19.0	< 0.0001	0.378	0.327	0.437	950	19.1	< 0.0001	0.590	0.498	0.699
not TP53 or CDKN2A not ANY RAS/MAPK	4174	25.8	< 0.0001	0.463	0.425	0.505	3444	21.7	< 0.0001	0.816	0.739	0.902

* Note that these P values indicate a significant inverse correlation between BRAF and ANY RAS/MAPK mutations in this cohort; such an inverse correlation is not present in the MSK-IMPACT cohort

REVIEWERS' COMMENTS

Reviewer #1 (Remarks to the Author):

In their revised manuscript, Darp et al. address each point/critique from reviewers with additional data and/or discussion to sufficiently clarify points of concern in the initial reviews.

Reviewer #2 (Remarks to the Author):

The authors have revised the manuscript in line with the Reviewers' comments, and I believe that the revision has strengthened the manuscript. The addition of the Mel-ST cells help demonstrate the biological relevance of the findings. The manuscript is now appropriate for publication in Nat Commun.

One comment: some of the data that the authors provided in their response to Comments #1 and #3 of my original review, should be included as Supplementary Information in the manuscript itself. While I appreciate that these are negative results, I think that they are still informative and could be of interest for readers.

In particular, I think it will be useful to include:

1) The results and a brief version of the discussion related to the MPS1 analysis (response to my comment #1b).

2) The results and a brief version of the discussion related to the absence of BRAFV600E-dependent activation of PLK1 (response to my comment #1d). I agree that the AURKAI results are impossible to interpret and should not be included in the revised paper.

3) Some of the results and a brief version of the discussion related to the lack of enrichment for BRAF or RAS/MAPK mutations in WGD tumors (response to my comment #3). At minimum, I think that the authors can add a paragraph to the Discussion, which would show and discuss (some) of the analysis that they performed. The key point being that BRAF or any RAS/MAPK mutations are not enriched in human melanomas that have undergone WGD, regardless of the p53 mutation status, suggesting that there are means other than BRAF or any RAS/MAP mutation by which tumors undergo WGD during melanoma tumorigenesis.

Reviewer #3 (Remarks to the Author):

As already stated in my original comments, the topic covered by this paper is of interest for a broad readership spanning from cell biologists to cancer researchers. The use of transgenic animal models combined with state-of-the-art imaging in human cell lines can be certainly seen as elements supporting publication in Nat Comms. All the criticism I raised in my original comments have been addressed thoroughly and the manuscript has substantially improved. Thus, the paper should be accepted without further delay.

POINT-BY-POINT RESPONSE TO REVIEWERS

We thank the reviewers for their comments. Below we address reviewer comments point-by-point. Changes in the manuscript based on reviewer comments are highlighted in yellow to facilitate any re-review.

Reviewer #1 (Remarks to the Author):

In their revised manuscript, Darp et al. address each point/critique from reviewers with additional data and/or discussion to sufficiently clarify points of concern in the initial reviews.

We thank the reviewer for helping us make the manuscript better and are pleased that our changes have been met with satisfaction.

Reviewer #2 (Remarks to the Author):

The authors have revised the manuscript in line with the Reviewers' comments, and I believe that the revision has strengthened the manuscript. The addition of the Mel-ST cells help demonstrate the biological relevance of the findings. The manuscript is now appropriate for publication in Nat Commun.

We thank the reviewer for the helpful comments and have addressed the additional suggestions as described below.

One comment: some of the data that the authors provided in their response to Comments #1 and #3 of my original review, should be included as Supplementary Information in the manuscript itself. While I appreciate that these are negative results, I think that they are still informative and could be of interest for readers.

In particular, I think it will be useful to include:

1) The results and a brief version of the discussion related to the MPS1 analysis (response to my comment #1b).

The data regarding MPS1 have been included in the new Supplementary Figure 3. In the main text we have included the following: "MPS1/TTK1, a kinase which is activated by BRAFV600E and promotes activation of the mitotic spindle checkpoint⁵⁰, was not activated in response to BRAFV600E expression in RPE-1 cells and thus not likely involved in BRAFV600E-driven cytokinetic failure (Supplementary Fig. 3B)."

2) The results and a brief version of the discussion related to the absence of BRAFV600E-dependent activation of PLK1 (response to my comment #1d). I agree that the AURKAi results are impossible to interpret and should not be included in the revised paper.

The data regarding PLK1 have been included in the new Supplementary Figure 3. In the main text we have included the following: "Polo-like kinase I (PLK1) helps to initiate cytokinesis⁴⁸, and it has been shown to interact with CRAF in G2/M⁴⁹. We tested whether cytokinesis failure was dependent on PLK1 and found that an inhibitor of PLK1 did not affect BRAFV600E tetraploid formation (Supplementary Fig. 3A)."

3) Some of the results and a brief version of the discussion related to the lack of enrichment for BRAF or RAS/MAPK mutations in WGD tumors (response to my comment #3). At minimum, I think that the authors can add a paragraph to the Discussion, which would show and discuss (some) of the analysis that they performed. The key point being that BRAF or any RAS/MAPK mutations are not enriched in human melanomas that have undergone WGD, regardless of the p53 mutation status, suggesting that there are means other than BRAF or any RAS/MAPK mutation by which tumors undergo WGD during melanoma tumorigenesis.

We have included a new Supplementary Figure 8 that contains data pertaining to BRAF and ANY RAS/MAPK mutations and WGD in human tumors. In the main text we have included the following: "Our analysis of human tumor samples confirmed that loss of p53 pathway activity was strongly correlated with WGD, including in tumors that harbor BRAF mutations and other mutations that activate RAS/MAPK signaling (Supplementary Fig. 8)." Additionally, we have included a brief statement in the Supplementary Figure 8 legend that helps readers interpret the findings, including the point that there are likely means other than BRAF or RAS/MAPK mutation by which tumors undergo WGD.

Reviewer #3 (Remarks to the Author):

As already stated in my original comments, the topic covered by this paper is of interest for a broad readership spanning from cell biologists to cancer researchers. The use of transgenic animal models combined with state-of-the-art imaging in human cell lines can be certainly seen as elements supporting publication in Nat Comms. All the criticism I raised in my original comments have been addressed thoroughly and the manuscript has substantially improved. Thus, the paper should be accepted without further delay.

We thank the reviewer for helping us make the manuscript better and are pleased that our changes have been met with satisfaction.